# Biochemical-free enrichment or depletion of RNA classes in real-time during direct RNA sequencing with RISER

Alexandra Sneddon [1,2,3], Agin Ravindran[1,2,3], Somasundhari Shanmuganandam[4,5], Madhu Kanchi[3], Nadine Hein [6], Simon Jiang[4,5,7], Nikolay Shirokikh [3] ✉ & Eduardo Eyras [1,2,3] ✉

The heterogeneous composition of cellular transcriptomes poses a major challenge for detecting weakly expressed RNA classes, as they can be obscured by abundant RNAs. Although biochemical protocols can enrich or deplete specified RNAs, they are time-consuming, expensive and can compromise RNA integrity. Here we introduce RISER, a biochemical-free technology for the real-time enrichment or depletion of RNA classes. RISER performs selective rejection of molecules during direct RNA sequencing by identifying RNA classes directly from nanopore signals with deep learning and communicating with the sequencing hardware in real time. By targeting the dominant messenger and mitochondrial RNA classes for depletion, RISER reduces their respective read counts by more than 85%, resulting in an increase in sequencing depth of 47% on average for long non-coding RNAs. We also apply RISER for the depletion of globin mRNA in whole blood, achieving a decrease in globin reads by more than 90% as well as an increase in non-globin reads by 16% on average. Furthermore, using a GPU or a CPU, RISER is faster than GPU-accelerated basecalling and mapping. RISER's modular and retrainable software and intuitive command-line interface allow easy adaptation to other RNA classes. RISER is available at https://github.com/comprna/riser.

Cellular transcriptomes encompass a diverse and unequally distributed range of RNA classes[1]. Consequently, highly abundant RNAs are over-represented in sequencing studies, countering the opportunity to characterize RNAs of low abundance. For example, long noncoding RNAs (lncRNAs) typically exhibit a low and highly tissue-specific expression pattern[2], which can be obscured by the dominant messenger RNA (mRNA) or mistaken for background noise in RNA abundance measurements[2]. Given the key roles of lncRNAs in multiple physiological processes, including development, neuronal function, and disease[3], it is critical to devise effective methods for their detection.

Targeted biochemical approaches have been critical for identifying novel lowly expressed RNA classes, notably lncRNAs[2,4]. Although effective in enriching or depleting specific RNA molecules prior to

[1]EMBL Australia Partner Laboratory Network at the Australian National University, Canberra ACT 2601, Australia. [2]Centre for Computational Biomedical Sciences, The John Curtin School of Medical Research, Australian National University, Canberra ACT 2601, Australia. [3]The Shine-Dalgarno Centre for RNA Innovation, The John Curtin School of Medical Research, Australian National University, Canberra ACT 2601, Australia. [4]Department of Immunity, Inflammation and Infection, The John Curtin School of Medical Research, Australian National University, Canberra ACT 2601, Australia. [5]Centre for Personalised Immunology, NHMRC Centre for Research Excellence, Australian National University, Canberra ACT 2601, Australia. [6]ACRF Department of Cancer Biology and Therapeutics, The John Curtin School of Medical Research, Australian National University, Canberra ACT 2601, Australia. [7]Department of Renal Medicine, The Canberra Hospital, Canberra ACT 2605, Australia. ✉e-mail: nikolay.shirokikh@anu.edu.au; eduardo.eyras@anu.edu.au

sequencing, biochemical targeting requires time-consuming and expensive specialized experimental protocols, which have been shown to induce RNA degradation and compromise the quality, length and content of the resultant reads[2,5,6]. They are also restricted in their applicability to a pre-determined set of transcripts. For instance, target cDNA capture using custom hybridization probes can uncover novel isoforms[4] but requires prior knowledge of the target regions. Similarly, mitochondrial RNAs (mtRNAs) have a distinct and well-characterized 3′ end that facilitates their specific targeting for depletion[5]. However, no approach supports the enrichment or depletion of any class of RNA without requiring the explicit definition of specific targets.

To enable the sensitive detection of lowly expressed RNAs without the limitations of biochemical treatment, we have developed RISER, a biochemical-free technology for the real-time enrichment or depletion of RNA classes. RISER seamlessly integrates with nanopore direct RNA sequencing (DRS), building on Oxford Nanopore Technologies' (ONT) read-until system, which allows the software to prematurely terminate the sequencing of individual molecules. RISER identifies RNA classes in real-time during sequencing, directly from just the first few seconds of raw nanopore signals using a deep learning model representing the target RNA class, and communicates with the sequencing hardware to physically eject unwanted RNAs from the pore, aiming to conserve sequencing capacity for the RNAs of interest.

As DRS proceeds from the 3′ ends of molecules, RISER exploits the common 3′ end properties of RNAs from the same class, which are assumed to be implicitly encoded at the start of the raw signal. For example, messenger RNAs (mRNAs) share common motif configurations in their 3′ untranslated regions (3′ UTRs)[7,8]. Without needing to model these features explicitly, RISER leverages the common signal patterns with a deep convolutional network to enable the detection of RNA classes, enabling targeted sequencing beyond the simple enumeration of sequence targets. Furthermore, RISER's direct signal classification strategy is more efficient, less computationally intensive, and can operate on shorter input lengths than using basecalling and mapping to a predefined list of sequence targets, and unlike real-time basecalling, RISER can be run with just a CPU.

Through testing using controlled datasets and live runs, we demonstrate that RISER can efficiently deplete multiple highly abundant RNA classes, subsequently increasing the read depth of low-abundance RNAs. By enabling the biochemical-free depletion of globin mRNA, RISER improves the efficiency of analysis of whole blood samples using long-read sequencing of native RNA. Extension to many RNA classes and straightforward adaptation to technology updates are facilitated by RISER's modular software design. Through a simple command-line tool, RISER empowers RNA researchers with a flexible and efficient strategy for biochemical-free targeted sequencing of native RNA.

## Results

### RISER identifies RNA classes from the 3′ ends of DRS signals

For the design of RISER (Fig. 1a), we prioritized both accuracy and real-time efficiency. An efficient approach is crucial since RNA molecules are typically shorter than DNA fragments sequenced in nanopore applications[9,10], while high accuracy is critical to ensure only molecules of interest are accepted through the pore, and none mistakenly rejected. We thus first considered the input signal length that would be short enough to allow assessment of the majority of molecules yet contain enough information for a correct decision. We found that signal lengths greater than 4 seconds (s), corresponding to approximately the first 280 nucleotides (nt) (R9.4.1 pore) of a transcript from the 3′ end, would lead to at least 17% of molecules escaping through the pore before a decision could be made (Fig. 1b) (Methods). We thus considered that for RISER to impact as many molecules as possible, a decision would need to be made within a maximum input length of 4 s.

To select RISER's model architecture, we initially considered the problem of separating mRNA and non-mRNA from the 3′ ends of DRS signals. Since DRS processes RNA molecules in the 3′ to 5′ direction, the initial portion of the nanopore signal always corresponds to the 3′ UTR for mRNAs or the 3′ end for non-coding RNAs. We hypothesized that differences in the molecular composition of the 3′ ends of mRNAs[7,8], which are implicitly encoded at the start of DRS signals, would make it possible to discriminate between mRNA and non-mRNA without the need for basecalling or mapping to a target reference.

We considered convolution-based architectures since they are ideally suited to capturing local temporal dependencies in time series inputs and can identify the relevant components for prediction irrespective of their location along an input signal[11]. These are key features for DRS signal analysis, given the variance in nanopore translocation speed and the local relationships between signal values for consecutive nucleotides. Position invariance is particularly crucial for this application since it is not known at which position in the 3′-end signal the relevant elements for prediction will be. Further, superior performance in terms of accuracy and efficiency of convolutional over recurrent architectures has been demonstrated for the analogous application of classifying species from nanopore DNA signals[12].

We trained and tuned three deep neural networks with convolutions that have shown strong performance in time series modeling tasks[11]: a residual neural network (ResNet)[13], a temporal convolutional network (TCN)[14], and a vanilla convolutional network (CNN). To enable efficient hyperparameter optimization in the development of the RISER architecture, we used a restricted number of datasets and DRS runs (Suppl. Table 1). Following hyperparameter optimization (Suppl. Tables 2–4), we tested the best-performing model for each candidate architecture on the binary classification of mRNA/non-mRNA. For testing, we considered input signal lengths of 4 s or less, in accordance with the 4 s maximum input length required for RISER's use in real-time applications.

The ResNet and CNN outperformed the TCN at all input lengths with respect to accuracy (Fig. 1c), area under the receiver operating characteristic curve (AUROC) (Suppl. Fig. 1a), and ratio of true (TPR) to false (FPR) positive rate (Fig. 1d), which better indicates the simultaneous maximization of accepted on-target molecules and rejected off-target molecules compared to the individual TPR, FPR, and precision metrics (Suppl. Fig. 1b–d). Importantly, the CNN was approximately twice as fast as both the ResNet and TCN (Fig. 1e) and was, therefore, selected as RISER's model (Fig. 1f).

### RISER integrates with the direct RNA sequencing platform

To deploy RISER in real-time during sequencing experiments, an intuitive command-line tool is provided for the user to select a target RNA class and a mode, enrich or deplete (Suppl. Note 1). Beneath the hood, RISER enacts targeted sequencing through ONT's ReadUntil application programming interface (API). This API allows third-party software to retrieve data from and send commands to individual pores in the sequencing hardware in real-time[15]. RISER continuously requests batches of in-progress sequencing signals from the API. For each signal received, RISER starts testing molecules after only 2 s of sequencing, after trimming the sequencing adapter and variable length poly(A) tail from the start of the signal (Suppl. Fig. 2) (Methods). If the predicted probability exceeds a tuned confidence threshold of 0.9 (Suppl. Fig. 3) (Methods), RISER sends a reject command to the sequencing hardware or allows the molecule to complete sequencing and stops requesting data for that molecule. If the probability is 0.9 or below, RISER tries to classify the molecule again when it is next received from the API after more of the molecule has transited (up to a maximum length of 4 s). If after 4 s a confident prediction has not been made, sequencing will continue unaffected, i.e., the RNA is let through the pore. By testing signals in this incremental manner, RISER is tolerant to variance in the RNA translocation speed.

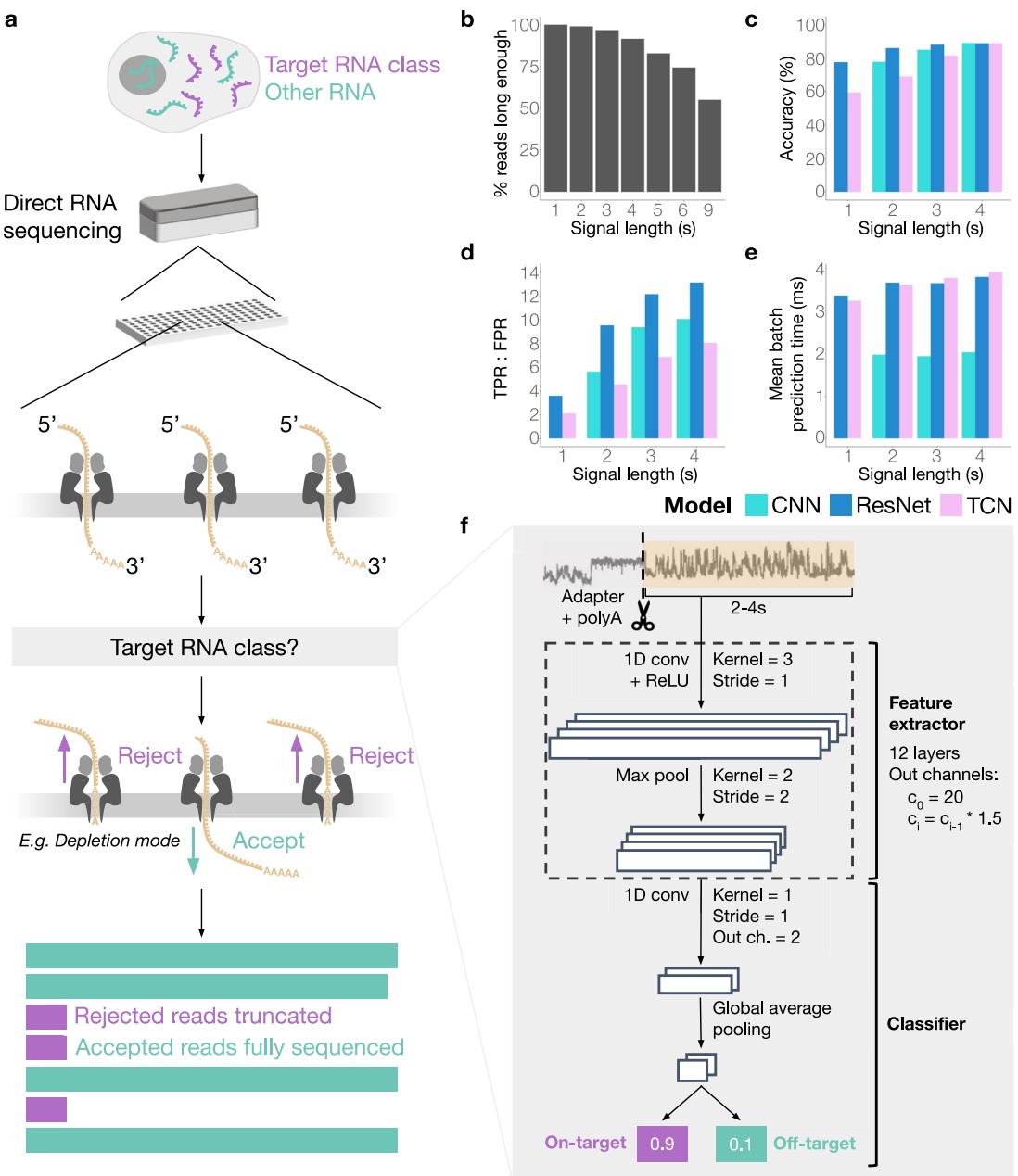

**Fig. 1 | RISER identifies RNA classes from the 3' end of DRS signals. a** RISER classifies RNA molecules as they commence sequencing by directly assessing raw nanopore signals, then sends an accept or reject decision to the sequencing hardware depending on the user-defined target RNA class and whether the user wants to enrich or deplete the target class (shown: target depletion). The accepted reads are sequenced to completion, while the rejected reads are truncated. **b** Percentage of reads in the training dataset (*y*-axis) with raw signals long enough to be input to RISER for each candidate input signal length expressed in seconds (*x*-axis). **c–e** Model performance on the test set for each candidate input signal length (*x*-axes), color-coded by the three convolutional network architectures assessed: vanilla convolutional neural network (CNN) (cyan), residual network (ResNet) (dark blue), temporal convolutional network (TCN) (pink). We show the accuracy (**c**), the ratio of true positive rate (TPR) to false-positive rate (FPR) (**d**), and the mean prediction time per batch of signals, expressed in milliseconds (**e**). **f** Neural network architecture for the CNN model selected to implement RISER. Source data for **b–e** are provided as a Source Data file.

For an initial test of RISER's integration with the sequencing platform, we utilized the playback feature of ONT's MinKNOW software, which allows signals recorded from a previous sequencing run to be replayed as though they were being generated in real-time[15]. We replayed a sequencing run from a cancer cell line (REH) that had not been used for model development or evaluation. In this simulated live-sequencing environment, when a reject command is issued, the signal being replayed is prematurely terminated; hence, read length provides an indirect measurement of accuracy. As expected, the RNA class targeted for enrichment showed significantly longer read lengths than their off-target counterparts (Suppl. Fig. 4).

## RISER is more efficient than sequence-based adaptive sampling

We compared the efficiency of RISER with sequence-based adaptive sampling (AS) such as that provided by ONT's MinKNOW software. Although both RISER and AS utilize the ReadUntil API for streaming data from and sending commands to the sequencing device, they critically differ in the signal-processing approach used to identify molecules. RISER performs a binary classification of the raw signal for a given RNA class, while AS basecalls and maps in-progress reads to a predefined list of target sequences.

We first compared the speed of AS and RISER for identifying mRNA in a test set of 1000 fixed-length signals. As in the MinKNOW AS

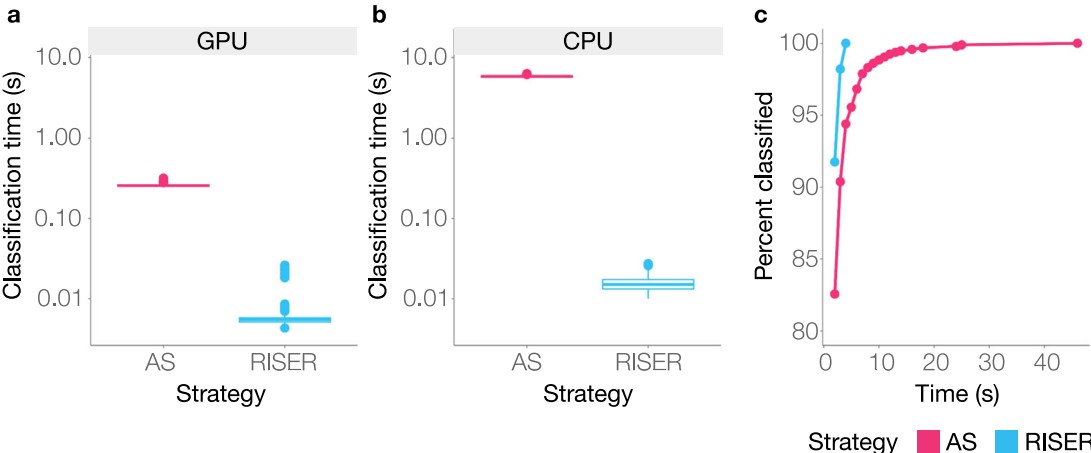

**Fig. 2 | RISER is more efficient than sequence-based adaptive sampling.**
**a**, **b** Time in seconds (*y*-axis, log10-scale) taken to classify fixed-length DRS signals (*n* = 1000 DRS reads randomly sampled from the mRNA test set) by sequence-based adaptive sampling (AS) using basecalling and mapping to the protein-coding transcriptome (fuchsia) and by classification with RISER's mRNA model (blue) using a GPU (**a**) or CPU (**b**). In the box plots, the lower and upper boundaries of the box are the first and third quartiles, with the median annotated with a line inside the box. The whiskers extend to the maximum and minimum values within 1.5 times the interquartile range. **c** Percentage of mRNA DRS signals classified as mRNA (*y*-axis) within a given time (*x*-axis) by AS (fuchsia) and RISER (blue). Source data are provided as a Source Data file.

implementation, the Dorado basecalling server[16] was used for basecalling and mapping to a reference listing all mRNA sequences in the human transcriptome, while RISER was tested using the mRNA model. Remarkably, when using GPU acceleration for both strategies, RISER was, on average, 44× faster than AS (Fig. 2a, Suppl. Table 5). When the same comparison was performed in CPU mode, RISER was, on average, 371× faster than AS (Fig. 2b, Suppl. Table 5). Notably, RISER in CPU mode was 16× faster than AS in GPU mode, while AS was prohibitively slow to be used in real-time with CPUs, with basecalling and mapping taking 5.8 s on average (Suppl. Table 5).

We next assessed the input signal length needed to identify targets using AS or RISER. This time, the objective was to correctly identify as mRNA a test set of 1000 fixed-length signals extracted from mRNA molecules. For a fair comparison, we considered 2 s as the minimum input length from which both technologies can make a prediction and incrementally increased the input length by 1 s until a prediction was made. Considering the signals that both technologies correctly identified as mRNA, RISER was able to identify all signals as mRNA within 4 s, while AS took up to 46 s to identify all of them (Fig. 2c). Thus, under comparable conditions, RISER is substantially faster, can be run using only CPUs, and requires less amount of signal to identify the desired targets compared to sequence-based adaptive sampling using basecalling and mapping, hence showing strong potential for the efficient control of live sequencing runs.

## RISER enables real-time depletion of abundant RNA classes

To test RISER in live sequencing runs, we considered the problem of depleting highly abundant RNAs, which occupy the majority of the sequencing capacity and obscure the detection of less abundant RNAs. Since mRNA is the most abundant class in standard poly(A)⁺ DRS runs (Suppl. Fig. 5), we first developed a model to target mRNA, intending to use it for depletion in live runs. To maximize generalizability, to train this model we used a larger training dataset (Suppl. Table 6) than previously used for the selection of RISER's architecture (Suppl. Table 1). Testing on a dataset from an independent cell line (HeLa) that was not used for training, hyperparameter tuning, or architecture selection, the mRNA model achieved high accuracy (94%), precision (0.99), and TPR (0.94), with low FPR (0.05) (Fig. 3a). We also considered the mRNA model's performance on individual RNA biotypes. Importantly, our model detected mRNA with high accuracy (94%) and was able to identify 99% of mtRNAs and 64% of lncRNAs as non-mRNAs (Fig. 3b).

To demonstrate the broad applicability and ease of training RISER for other targets, we next developed a second model to identify mtRNAs, which are highly abundant in RNA sequencing of cardiac and other muscle samples (30-80% of reads[5]) (Suppl. Fig. 5). Evaluation of the mtRNA model on our independent HeLa dataset demonstrated high accuracy (99%), precision (93%) and TPR (98%), while maintaining a low FPR (1%) (Fig. 3c). Our mtRNA model was also able to correctly classify as non-mtRNA all RNA biotypes with >98% accuracy (Fig. 3d). Furthermore, the performance of the mRNA and mtRNA models was recapitulated on a separate independent cell line experiment from a different lab[17], demonstrating consistent performance despite a different sample source and sequencing location (Fig. 3e–h).

Given the strong performance of RISER's mRNA and mtRNA models in non-live independent experiments, we next explored their utility for real-time depletion in live sequencing runs. Live sequencing of a standard DRS library from HEK293 cells was conducted using a MinION Mk1B, with the flow cell channels split into two groups to simultaneously test RISER depleting both mRNA and mtRNA and no RISER as a control, thus avoiding any possible flow cell biases. RISER was executed as described above, using both mRNA and mtRNA models to predict the RNA class (Methods).

Consistent with our expectation that RISER prematurely truncates the reads in the RNA classes targeted for rejection, we found that RISER significantly reduced the length of the reads from both target classes relative to the control condition (Fig. 4a). We also detected a smaller but significant difference in the read lengths in lncRNAs. However, we found that this level of variability is expected for lncRNAs (Suppl. Fig. 6). Moreover, demonstrating that the RISER decision is effectuated within 4 s of sequencing, the read coverage per base in individual mRNA and mtRNA transcripts markedly dropped off within 280nt (~4 s) upstream of the transcripts' 3′ end when RISER was used for target depletion (Fig. 4b, upper & middle panels). In contrast, the lncRNA coverage in the control and deplete conditions remained similar (Fig. 4b, lower panel), as expected for RNAs not targeted for depletion. In agreement with these observations, RISER also substantially reduced the transcript fraction covered by the sequenced reads across all transcripts in the mRNA and mtRNA target classes, whereas the covered transcript fraction for lncRNAs remained the same with or without RISER (Fig. 4c).

We next quantified RISER's impact on sequencing depth, finding that RISER reduced the read counts for transcripts in the mRNA and

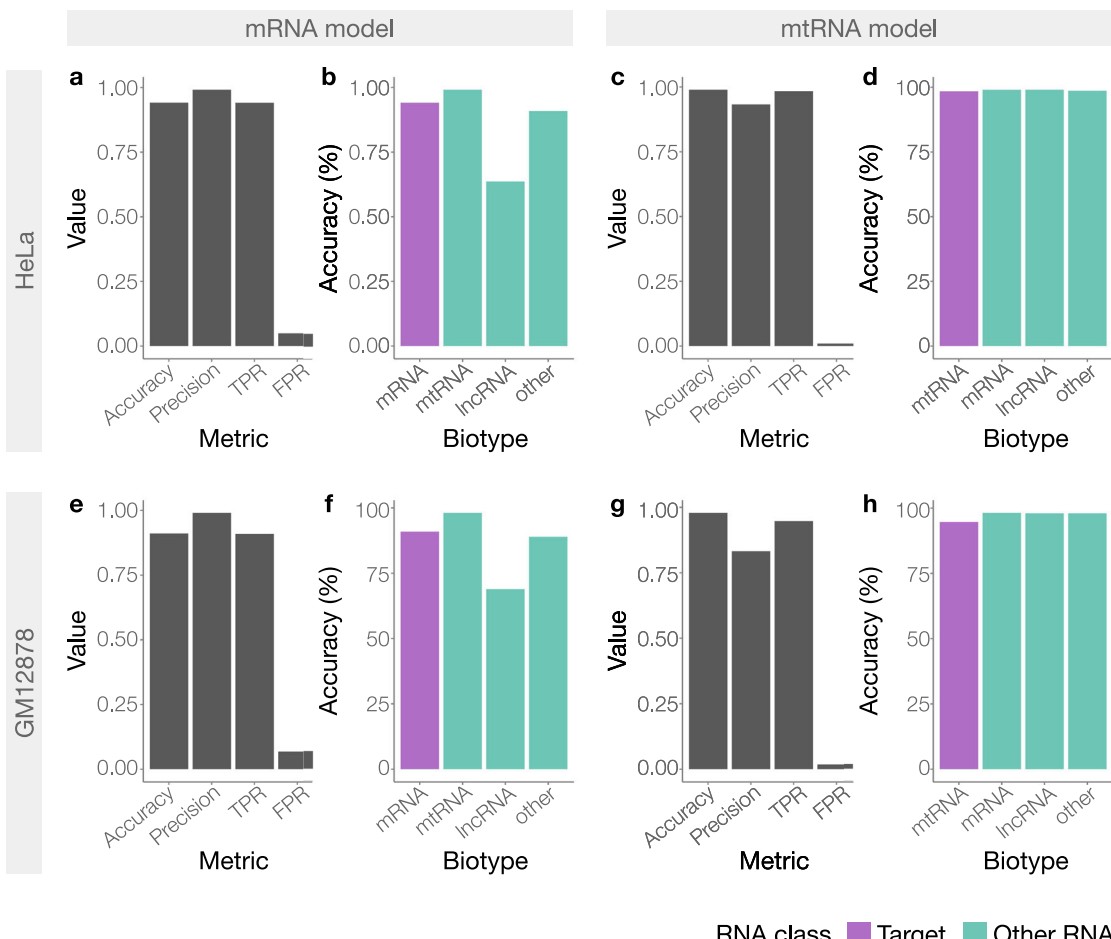

**Fig. 3 | Performance of RISER models for the depletion of mRNA and mtRNA.** Performance in non-live independent experiments, using poly(A)$^+$ RNA from HeLa cells (**a–d**) and GM12878 cells (**e–h**). For the mRNA (**a, e**) and mtRNA (**c, g**) models in each experiment, we show overall accuracy, precision, true positive rate (TPR) and false positive rate (FPR). For the same mRNA (**b, f**) and mtRNA (**d, h**) models, we show the accuracy for each biotype, color-coded by whether the biotype belongs to the class targeted for RISER depletion (purple) or not (teal). Source data are provided as a Source Data file.

mtRNA target classes by an average of 86% ($n = 873$) and 85% ($n = 14$), respectively (Fig. 4d). Remarkably, lncRNA transcripts had an increased read count of 23–93% (47% on average, $n = 8$) (Fig. 4d). Considering sequenced nucleotides, RISER led to an 89% and 90% reduction in average nt counts for mRNA and mtRNA transcripts, respectively, as well as an increase in nt counts by 49% on average for lncRNA transcripts (Fig. 4e). By comparing with the percent change in read or nt counts between control runs, we found that the effect of RISER was statistically significant for all three biotypes (Fig. 4d, e). The significant impact of RISER on read and nucleotide counts was recapitulated using a second control experiment (Suppl. Fig. 7). Considering RISER's decision per molecule in each target RNA class, 88% of mRNA and 98% of mtRNAs were correctly rejected (Suppl. Fig. 8a). RISER erroneously accepted 5% of mRNAs and 1% of mtRNAs, while the remainder were not detected with sufficient confidence.

We systematically analyzed the RISER model errors to identify any possible biases toward specific transcripts. Of the 2902 unique mRNA transcripts identified in the RISER condition, only 6 were sequenced to completion for at least 50% of their copies (Suppl. Fig. 8b), while none of the 14 mtRNA transcripts were (Suppl. Fig. 8c). Of the 52 unique lncRNA transcripts sequenced, RISER erroneously rejected 5 of them for at least 50% of their copies (Suppl. Fig. 8d). Four of these five had common sequences with protein-coding transcripts, making it not possible for RISER to distinguish them from mRNA. This suggests that errors in the remaining 99.99% of mRNA transcripts, 90% of lncRNA transcripts, or any mtRNA transcript may stem from sequencing noise.

Furthermore, we found that by depleting mRNA and mtRNA, RISER did not impact the relative abundance of lncRNAs, with the correlation of lncRNA relative abundances between RISER and control conditions (Pearson $R = 0.75$, $p = 8.4E-11$) (Suppl. Fig. 9a) consistent with the correlation of lncRNA relative abundances between independent HEK293 sequencing experiments without RISER (Pearson $R = 0.76$, $p = 2.6E-6$) (Suppl. Fig. 9b). Moreover, while the final percentage of available pores varied generally between different runs after 24 h, these numbers were similar between RISER and control experiments (Suppl. Fig. 10a–c).

## RISER enables biochemical-free depletion of globin mRNA in whole blood samples

To further demonstrate RISER's broad utility, we next applied it for the depletion of mRNA originating from globin genes, which makes up to 80–90% of the read counts in whole blood short-read RNA sequencing experiments[6,18] and around 60% in DRS experiments (Fig. 5a). Currently, there is no method available for globin depletion that is compatible with DRS. We hypothesized that conserved regulatory sequence motifs in the 3′ end of globin mRNAs, such as the pyrimidine-rich elements that contribute to globin mRNA stability necessary for ample hemoglobin production[19], would be implicitly encoded in DRS signals and therefore exploitable by a RISER model. We thus trained a RISER model for globin mRNA identification (Methods). Testing on a reserved dataset of DRS reads from whole blood, this model achieved high accuracy (98%), precision (99%), TPR (98%), and low FPR (0.5%)

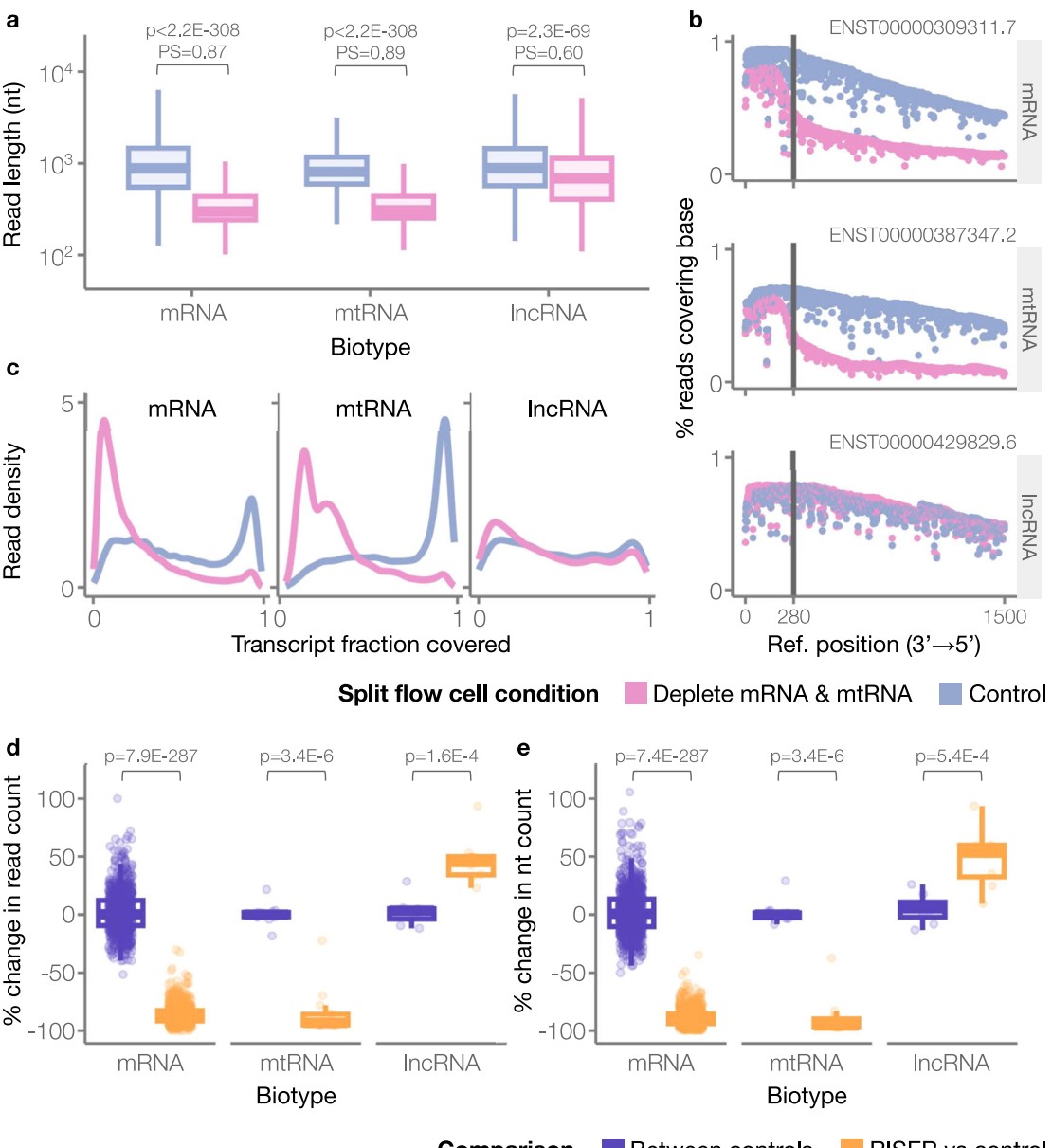

**Fig. 4 | RISER enables real-time depletion of mRNA and mtRNA.** RISER performance during live sequencing of poly(A)⁺ RNA from HEK293 cells, using a MinION Mk1B flow cell split into two conditions: RISER targeting both mRNA and mtRNA for depletion (pink), and no RISER as a control (blue). **a** Distribution of read lengths (*y*-axis, log10-scale) for each RNA class in each condition. In the box plots, the lower and upper boundaries of the box are the first and third quartiles, with the median annotated with a line inside the box. The whiskers extend to the maximum and minimum values within 1.5 times the interquartile range. Outliers were not included. The read lengths of each RNA class were compared in the control and deplete conditions using a one-tailed Wilcoxon rank sum test (H1: control > deplete). The probability of superiority (PS) is also shown above each comparison. PS is the probability that a randomly sampled read from the control condition is longer than a randomly sampled read from the deplete condition (i.e., PS close to 0.5 means the lengths are likely to be the same, whereas PS close to 1 means that the control lengths are highly likely to be larger) (mRNA: *p*-value $p < 2.2E-308$, test statistic $U1 = 30493942144.5$, $PS = 0.87$ and $n = 376,048$, mtRNA: $p < 2.2E-308$, $U1 = 909552316.0$, $PS = 0.89$ and $n = 65,569$, lncRNA: $p = 2.3E-69$, $U1 = 13787658.0$, $PS = 0.60$ and $n = 9574$). **b** Percentage of reads covering the first 1500 bases from

the 3′ ends of the transcript (*y*-axis) for an example mRNA (upper panel), mtRNA (middle panel) and lncRNA (lower panel). The reference positions (*x*-axes) are ordered from 3′ to 5′. The vertical line indicates 280nt upstream of the 3′ end, which approximately corresponds to the maximum RISER input length of 4 s. **c** Density distributions of the transcript fraction (*x*-axis) covered by the sequenced reads. **d**, **e** Distribution of the percent change in read (**d**) and nucleotide (nt) (**e**) counts (*y*-axis), with respect to a control run, when RISER was used to deplete mRNA and mtRNA (orange) and for a separate control run (purple). In the RISER vs control comparison, lncRNA read counts increased from 602 (control) to 885 (depletion of mRNA and mtRNA by RISER). The box plots are defined the same as in (**a**). Outliers were not included. For the set of transcripts in each biotype, the percent change in nt or reads using RISER was compared to the (no-RISER) control using a paired one-tailed Wilcoxon signed rank test (H1 for mRNA and mtRNA: RISER vs control < between controls, H1 for lncRNA: RISER vs control > between controls). For this comparison, lncRNAs that did not overlap with coding exons from protein-coding transcripts were used. Test parameters and statistics are provided in Suppl. Table 8. Source data are provided as a Source Data file.

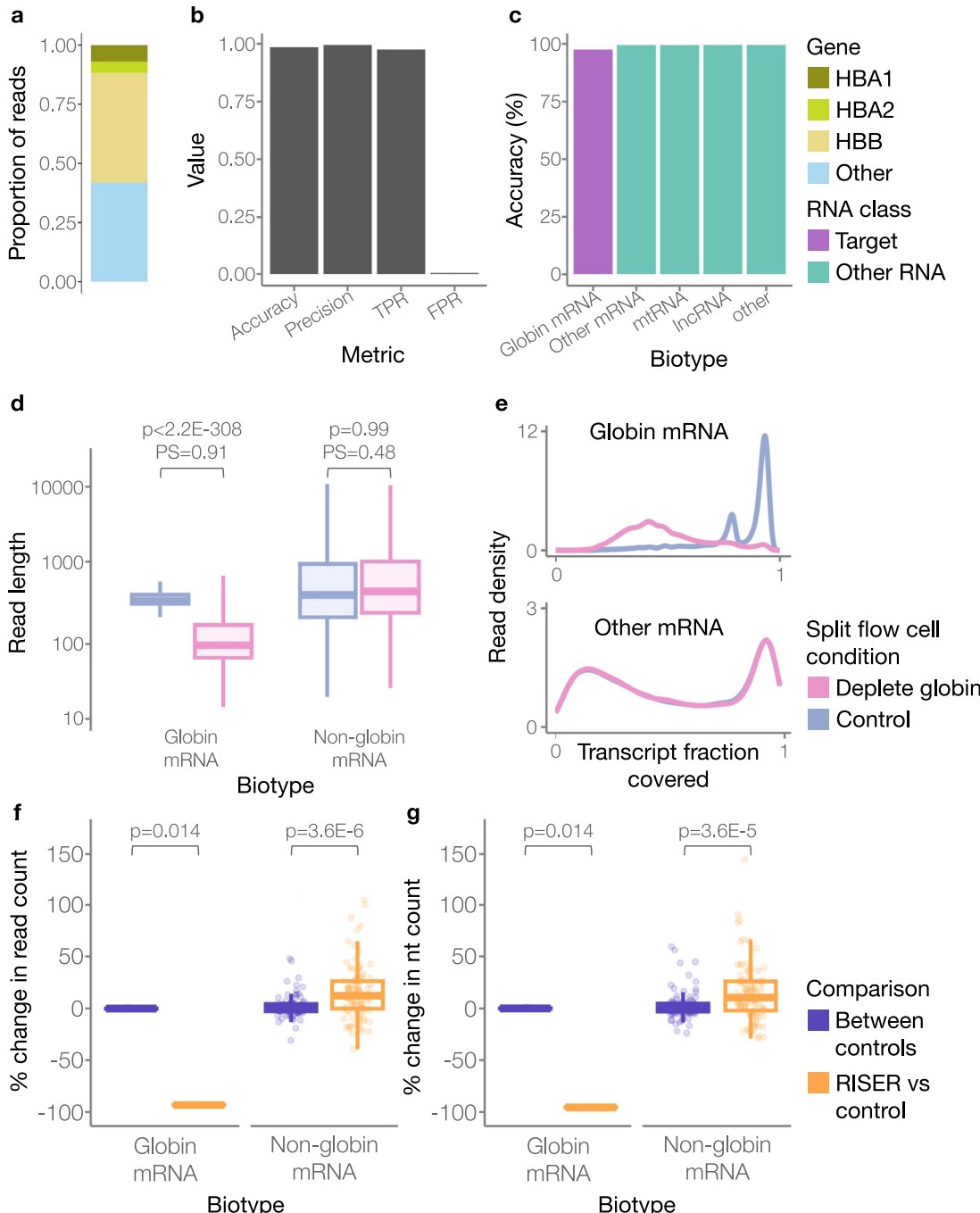

**Fig. 5 | RISER enables real-time depletion of globin mRNA. a** Proportion of reads originating from globin genes (*HBB*, *HBA1*, *HBA2*) or other genes in a standard DRS run of a whole blood sample. **b** Performance of the RISER model for the detection of globin mRNA using DRS reads from a non-live experiment of whole blood. We show the performance metrics of accuracy, precision, true positive rate (TPR) and false positive rate (FPR). **c** Accuracy of the globin mRNA model per biotype, color-coded by whether the biotype belongs to the class targeted for RISER depletion (purple) or not (teal). **d** Distribution of read lengths (*y*-axis, log10-scale) for globin mRNAs and non-globin mRNAs in each condition. In the box plots, the lower and upper boundaries of the box are the first and third quartiles, with the median annotated with a line inside the box. The whiskers extend to the maximum and minimum values within 1.5 times the interquartile range. Outliers were not included. The read lengths of each RNA class were compared in the control and deplete conditions using a one-tailed Wilcoxon rank sum test (H1: control > deplete). The probability of superiority (PS) is also shown. PS is the probability that a randomly sampled read from the control condition is longer than a randomly sampled read from the deplete condition (PS close to 0.5 means the lengths are likely to be the same,

whereas PS close to 1 means that control lengths are highly likely to be larger) (globin mRNA: $p < 2.2\text{E}{-}308$, $U1 = 30726328836.5$, $PS = 0.91$ and $n = 391,906$, non-globin mRNA: $p = 0.99$, $U1 = 144441871.5$, $PS = 0.48$ and $n = 34,848$). **e** Density distributions of the transcript fraction (*x*-axis) covered by the sequenced reads for globin (upper panel) and for non-globin (lower panel) mRNA. **f, g** Distribution of the percent change in read (**d**) and nucleotide (nt) (**e**) counts (*y*-axis), relative to a control run, when RISER was used to deplete globin mRNA (orange) and for a separate control run (purple). In the RISER vs control comparison, non-globin mRNA read counts increased from 6776 (control) to 7363 (depletion of globin mRNA by RISER). The box plots are defined the same as in (**d**). Outliers were not included. For the set of transcripts per biotype, the percent change in nt or reads using RISER was compared to the (no-RISER) control using a paired one-tailed Wilcoxon signed rank test (H1 for globin mRNA: RISER vs control < between controls, H1 for non-globin mRNA: RISER vs control > between controls). Test parameters and statistics are provided in Suppl. Table 8. Source data are provided as a Source Data file.

(Fig. 5b). Critically, our globin model was able to correctly detect globin mRNA with 98% accuracy and to correctly classify all other RNA biotypes as non-globin with >99% accuracy (Fig. 5c).

Compelled by the model's strong performance in controlled tests, we next tested the efficiency of globin mRNA depletion using RISER during live MinION sequencing of a standard DRS library from whole blood. As expected, RISER significantly reduced the length of globin mRNA reads in the deplete compared to the control condition, while the lengths of other mRNAs remained unaffected by RISER (Fig. 5d). RISER also reduced the transcript fraction covered by the sequenced reads in the globin mRNA, whereas the covered transcript fraction for other mRNAs remained the same with or without RISER (Fig. 5e). Importantly, RISER reduced the read count of each globin mRNA transcript by 93% on average ($n = 4$), leading to an increase in reads for non-globin mRNA transcripts of 16% on average ($n = 100$) (Fig. 5f). Considering sequenced nucleotides, RISER led to a 96% average reduction of nt counts for globin mRNA transcripts and a 14% average increase for non-globin mRNA (Fig. 5g). Compared to the change in read counts between two randomly selected groups in an independent control experiment, the changes in nt and read counts achieved by RISER were statistically significant (Fig. 5f, g). The statistical significance of RISER's effect was recapitulated considering a second independent control experiment (Suppl. Fig. 11). While we observed similar trends in the pores available between the RISER and control conditions across the 24 h sequencing runs, the final number of available pores varied between each run (Suppl. Fig. 10d–f). Considering RISER's prediction per molecule, a correct decision was made for at least 98% of molecules in each RNA class (Suppl. Fig. 12a). Unpacking the few errors in each RNA class to determine if the model was biased toward specific transcripts, we found that most of the molecular copies of all the eight globin mRNA transcripts were correctly classified (Suppl. Fig. 12b), while every mtRNA was correctly classified as non-globin at least 95% of the time (Suppl. Fig. 12c), suggesting no bias within either of these classes. Only one mRNA transcript that was not in our globin mRNA category (containing *HBA1*, *HBA2*, and *HBB* transcripts) was consistently classified as globin mRNA (Suppl. Fig. 12d). Intriguingly, this belonged to the hemoglobin subunit delta (*HBD*) gene, which was not included in the model due to an insufficient number of reads available for training. Despite not being seen during training, the model generalized to identify mRNA molecules from the *HBD* gene as belonging to the globin class, indicating the model has learned to identify general features of globin mRNAs. Only one lncRNA was sequenced, and it was correctly identified as non-globin in 12 of the 13 molecular copies. Finally, we observed that the correlation between relative abundances of non-globin mRNAs was conserved between the RISER and control conditions (Pearson $R = 0.9$, $p < 2.2E{-}16$) (Suppl. Fig. 13a), consistent with the observed correlation between separate whole blood sequencing experiments performed without RISER (Pearson $R = 0.92$, $p < 2.2E{-}16$) (Suppl. Fig. 13b).

## Discussion

RISER leverages the read-until functionality of ONT sequencing platforms to focus the limited operating time of the nanopores on a user-defined RNA class of interest, enabling biochemical-free targeted sequencing of native RNAs. By combining targeted depletion or enrichment with DRS, RISER enables downstream analysis in nanopore signal space, such as the study of RNA modifications[20]. Targeted sequencing with any other sequencing technology is limited by the inability to measure RNA in its native state, which precludes studies of the epitranscriptome and risks the introduction of biases that typically result from reverse transcription.

A key feature of RISER is that it performs the reject decision by directly classifying the raw signal rather than using real-time base-calling and mapping as in ONT's sequence-based adaptive sampling. We have shown that RISER's approach is substantially faster, even

when run on a CPU, and requires shorter input lengths to identify targets than GPU-accelerated AS. This confers RISER with an advantage for targeted DRS, as a highly efficient targeting strategy is crucial given the overall shorter lengths of RNA molecules compared to sequenced DNA fragments[9,10]. Although strategies have been developed to optimize AS by accelerating the mapping step[21] or making decisions from the signal directly[22,23], they have not been applied to DRS.

A second key feature of RISER is that it defines the target as a class represented by a deep neural network model rather than as a list of pre-defined specific sequences, as in AS. Unlike RISER, the latter approach precludes the opportunity to target novel transcripts, which could be of critical relevance for organisms that lack a well-annotated reference or have an insufficiently characterized transcriptome. Furthermore, since RISER does not require a specific input length to identify targets but rather assesses any signal between 2 and 4 s long, it can be applied to other RNA classes of various optimal input lengths without the need to change the signal processing approach.

We illustrated RISER's usability during live runs by targeting the highly abundant mRNA and mtRNA classes for depletion. While demonstrating the effective depletion of both targets, RISER also led to a moderate increase in sequencing depth for many lncRNAs. Although this was encouraging, we expect there is scope to improve these depth gains by increasing the lncRNA DRS datasets available for model training. At present, accurate lncRNA identification from the 3′ end signal remains challenging. As a more diverse, less abundant, and less well-defined class[24], lncRNAs are harder to detect compared to other RNA classes. They also frequently present cell type-specific expression, which makes it harder for predictive models to generalize to new samples. It is thus crucial to maximize the diversity and quantity of samples used for training. Despite these challenges, our mtRNA and globin mRNA models were able to differentiate lncRNAs with remarkably high accuracy.

Further demonstrating RISER's broad applicability and potential for impact in clinical settings, we applied it to the depletion of globin mRNA from blood samples. mRNAs originating from globin genes typically dominate whole blood sequencing runs, thereby consuming sequencing resources and reducing the read coverage of the other transcripts of interest in clinical and other diagnostic samples. Although accurate profiling of whole-blood transcriptomes is essential in medicinal and biological discovery, no globin depletion method exists that is compatible with DRS. Furthermore, bioinformatically discarding globin reads post-sequencing is insufficient to counteract the problem of limited coverage of non-globin RNAs[25]. Biochemical approaches to remove or enzymatically degrade globin mRNAs[6,18] are known to induce a higher coverage towards the 3′ end of transcripts, can result in frequent off-target depletion, and fragment the RNA[26]. These limitations make globin depletion only viable for certain types of analyses based on short-read sequencing technologies. Problematically, short-read methods require cDNA as input and are incapable of sequencing entire transcripts, requiring error-prone computational transcript reconstruction with probabilistic approaches[27]. Although alternative biochemical-based methods have been proposed that minimize degradation to enable long-read sequencing[26], they have limited efficiency, and are only compatible with cDNA sequencing. RISER thus has the potential to contribute to the development of efficient long-read direct RNA sequencing applications in blood samples.

There may be several avenues to explore the further optimization of RISER. Using larger datasets for training, especially for the lowly expressed RNA classes, could improve RISER's sensitivity to rare RNAs. Other architectures designed for sequential modeling of signals, with comparable efficiency to CNNs, may lead to improvements in RISER's approach. While models such as RNNs or transformers may yield accuracy gains, they are generally more computationally expensive than lightweight CNNs[14,28], as we have used in RISER, which is key to

targeted DRS. Other aspects of the technology may not be easily addressed as they depend on the ONT sequencing infrastructure. For instance, the lag between the read-until API receiving a reject command from RISER and requesting a voltage reversal in the hardware, as well as the efficiency of the voltage reversal itself, is beyond RISER's control and may vary across sequencing platforms. Moreover, there is a risk of RNA secondary structures forming on the trans side of the pore, which could impede the efficiency of or completely inhibit reverse translocation by voltage reversal. In any case, with new iterations of the sequencing technology providing higher throughput and less noisy signals, the potential to amplify the applicability and impact of RISER is enhanced.

Finally, RISER has been developed using best practices in software development. RISER is freely available to use through a simple and intuitive command-line tool (Suppl. Notes 1–3, Suppl. Table 7), with no requirements for additional files such as BAM or BED files. RISER's modular design (Suppl. Figs. 14 and 15) facilitates easy adaptation of each software component, such as the incorporation of new models or the extension to new iterations of the sequencing platforms. We also provide the software code to retrain the neural network for new RNA classes, thus enabling the application to other RNA classes or the identification of RNA from different organisms. In summary, RISER empowers researchers across multiple fields to perform efficient and cost-effective, real-time targeted sequencing of native RNA molecules, catalyzing a new generation of RNA enrichment targets and sequencing control.

## Methods

We confirm that our research complies with all relevant ethical regulations. Procedures were conducted in accordance with the Australian National University ethics protocol No. ETH.1.16.01/ETH.01.15.015.

### RISER model development

**DRS datasets used**. For the development of the RISER model, MinION DRS signals from human heart[29], GM24385 cells (sequenced in this study), and HEK293 cells[30,31] were used (Suppl. Table 1) and are hereafter collectively referred to as the *model development datasets*.

**GM24385 RNA extraction and sequencing**. The lymphoblastoid cell line (LCL; from the National Institute of General Medical Sciences (NIGMS) Human Genetic Cell Repository) GM24385 (Coriell Institute for Medical Research) was grown in RPMI1640 media (Gibco by Thermo Fisher Scientific; cat. No. 11875093) supplemented with 15% Hi-FCS (Gibco by Thermo Fisher Scientific; cat. No. 26140079) and 2 mM L-Glutamine (Thermo Fisher Scientific; cat. No. 25030149) in 6-well plates (Coning; cat. No. Z717266) under 5% $CO_2$. Cells were subcultured in a 1:3 split and were harvested at a cell density density of $1 \times 10^6$ cells/ml. Cell pellet collection was performed by transferring GM24385 cell suspension into 15 ml conical centrifuge tubes (Merck; cat. No. CLS430055) and centrifuging at $500 \times g$ for 10 min at room temperature.

To isolate RNA from the cytoplasmic and nuclei fractions, $10^7$ cells were lysed in 200 μl of the non-denaturing lysis buffer containing 25 mM HEPES-KOH (pH 7.6 at 25 °C), 50 mM KCl, 5.1 mM $MgCl_2$, 2 mM DTT, 0.1 mM EDTA, 5% v/v glycerol, 2× Complete EDTA-free protease inhibitor and 0.5% v/v Igepal CA-630B. Cells were resuspended in the lysis buffer through pipetting, and RNasin Plus (Promega; cat. No. N2611) was immediately added to the final concentration of 1 U/μl. Cell lysis was completed by passing the lysate 4× through a 20-gauge needle followed by passing it 4× through a 27-gauge needle attached to a 1 ml syringe. The cell lysate was then centrifuged at $1000 \times g$ for 5 min at 4 °C. The supernatant was transferred into a new 1.5 ml tube and mixed with 350 μl of the RA1 lysis buffer, followed by RNA isolation using silica columns (NucleoSpin RNA kit; Macherey Nagel; cat. no.

740955.50) to obtain the cytoplasmic RNA fraction. The process included on-column DNase treatment with TURBO DNase (Thermo Fisher Scientific; cat. no. AM2238). The pelleted nuclei were resuspended in 1 ml of ice-cold sterile PBS (Sigma-Aldrich; cat. No. 9408-64-8) and 50 μl counted using a cell counter (Beckman-Coulter). The nuclei suspension was spun again at $1000 \times g$ for 5 min at 4 °C, the supernatant aspirated, and $6.6 \times 10^6$ nuclei were lysed in 700 μl of the RA1 RNA lysis buffer and isolated using the silica columns as described above. RNA isolated from the cytoplasmic fraction was eluted from the columns in 80 μl and from nuclei in 120 μl of RNase-free water and stored at −80 °C.

For in vitro polyadenylation, ~9 μg of the RNA in 94 μl of deionized water or 25 mM HEPES-KOH (pH 7.6 at 25 °C), 0.1 mM EDTA (HE) buffer were first denatured by incubating at 65 °C for 3 min and immediately chilling in ice. The solution was then supplemented with 12 μl of 10× *E. coli* Poly(A) Polymerase buffer (New England Biolabs), 8 μl of 1 mM ATP and mixed. To the resultant solution, 3 μl of 40 U/μl RNasin Plus (Promega) and 3 μl of 5 U/μl *E. coli* Poly(A) Polymerase (New England Biolabs; cat. no. M0276S) were added and mixed, and the resultant mixture incubated at 37 °C for 30 min. The eluate from in vitro polyadenylated RNA was further purified following an established protocol for RNA cleanup with SPRI beads[30] (AMPure XP, Beckman-Coulter; cat. No. A63882).

DRS libraries were constructed using the RNA kit supplied by Oxford Nanopore Technologies (cat. No. SQK-RNA002), generally according to the manufacturer's recommendations with minor amendments. 3000–1500 ng of RNA were used for each 2× library preparation within every replicate (with all recommended volumes doubled up) with a direct RNA sequencing kit (SQK-RNA002), as supplied by Oxford Nanopore Technologies. The modifications were that Superscript IV RNA Polymerase (Thermo Fisher Scientific) was used, RNA Control Standard (RCS) was omitted, and RNasin Plus (Promega) was included at 1 U/μl in all reaction solutions up until the reverse transcription reaction in the SQK-RNA002 sequencing protocol. The final adapter-ligated sample was eluted in 40 μl. Two DRS runs for the nuclei RNA libraries and one DRS run for the cytoplasmic RNA library were conducted on a MinION Mk1B equipped with R9.4.1 flow cells (Oxford Nanopore Technologies; cat No. FLO-MIN106D) for 44 h, 29 h, and 72 h, respectively. Initially, the flow cells were left at 25 °C for 30 min to reach ambient temperature. The flow cells were then inserted into the MinION Mk1B, and a quality check was performed to ensure that the pore count was above the manufacturer warranty level (800 pores). Prior to sample loading, the priming solution (Flush Buffer + Flush Tether) was degassed in a vacuum chamber for 5 min. A similar approach was used when loading the RNA library. The run set-up on the loaded libraries was performed using the default MinKNOW software (Version 20.10.3) run configuration. The SQK-RNA002 sequencing option was selected. For real-time assessment of the quality of the run, the output FAST5 files were basecalled in line with sequencing using Guppy in 'fast' basecalling mode.

**Input signal length selection**. The input signal length to use for the RISER model was determined by considering the trade-off between needing to minimize input length for efficient enrichment or depletion while also ensuring the signal contained enough information for a correct prediction. This trade-off was assessed by observing the lengths of the transcript signal portion (i.e., the part of the signal that RISER predicts, which is the signal remaining after the sequencing adapter and poly(A) tail have been removed) for the signals in the model development datasets.

First, the sequencing adapter and poly(A) tail were removed from the raw nanopore signals using BoostNano[32]. Next, the percentage of signals that had a transcript signal length at least as long as the candidate input signal lengths of {1–6,9} s was calculated. The transcript signals that were shorter than each candidate's input signal length

corresponded to molecules that would be too short and, therefore, escape through the pore before a decision could be made. The longest signal length that would still allow at least 90% of molecules to be assessed was selected as the maximum input length to use for RISER, which was 4 s.

**Train and test data preparation.** To curate the training and testing sets for developing the RISER model, the reads in the model development datasets were basecalled, mapped, and filtered as follows[30]. Fast5 files were basecalled with Guppy (v4.0.14) using options *–flowcell FLO-MIN106 –kit SQK-RNA002* and mapped to the GENCODE reference transcriptome (release 34, assembly GRCh38.p13) with minimap2[33] (v2.17) using options *-ax map-ont --secondary=no -t 15*. To retain only high-confidence mappings, alignments to the reverse strand, secondary or supplementary mappings and unmapped reads were removed using samtools[34] (v1.10) with option *-F 2324*.

Using the GENCODE (release 34, assembly GRCh38.p13) transcript biotypes, reads were then split by biotype into mRNA and non-mRNA (all other biotypes) classes, with pseudogenes removed to ensure there were no common sequences between the two classes. For the model training and testing datasets, each class was further split by randomly dividing the reads into 80% training and 20% testing. Twenty percent of the training data was reserved for hyperparameter tuning. To resolve class imbalance, the majority class (mRNA) was undersampled to achieve a 50/50 class balance in each of the train, test, and tune sets so that the model was trained in an unbiased way.

The maximum RISER input length of 4 s was used for model training and testing to maximize the amount of information learned by the model. After removing the sequencing adapter and poly(A) tail from the start of each raw nanopore signal with BoostNano[32] (commit 8715800), the first 4 s (s) of the remaining transcript signal was then extracted and normalized using median absolute deviation with outlier smoothing. Signals with transcript signal lengths less than 4 s were discarded. The final training set contained 1,073,720 signals.

**Candidate neural network architectures.** Convolutional neural networks capture spatial structure in the input by using convolutions, which are computed as the dot product between a filter (also known as a *kernel*, which is a matrix of learnable weights) and a portion of the input the same size as the filter (a *local receptive field*). The filter acts as a feature detector, and by *sliding* the same filter across the input to produce a feature map, feature detection becomes translation invariant, i.e., the same feature can be found anywhere along the input length. The use of multiple filters (*channels*) in each layer allows different features to be detected, while the use of multiple layers in the network allows hierarchies of features to be learned[35].

Three variants of convolutional architectures known to have strong performance in 1D sequence modeling tasks were considered: the Residual Neural Network (ResNet)[13], the Temporal Convolutional Network (TCN)[14], and a vanilla convolutional network (CNN). For each architecture, the hyperparameter configuration was systematically tuned. All models were trained using binary cross-entropy loss and Adam optimization for up to 100 epochs (within a 48-hour time limit), after which their accuracy was evaluated on the validation set (Suppl. Tables 2–4). All models were built, trained, and tested using PyTorch (v1.9.0)[36] with a single NVIDIA Tesla V100 graphics processing unit (GPU).

**Residual network (ResNet) hyperparameter optimization.** The ResNet architecture overcomes convergence issues when training deep networks[13] by using shortcut connections that directly propagate unmodified inputs to subsequent layers. The effect is a reduced backpropagation distance to mitigate gradient update instability, enabling the training of much deeper networks and the extraction of richer feature hierarchies than was previously possible[13].

Thirty-three variants of the following general ResNet architecture were trained and tested; the input vector was fed into a feature extractor layer composed of a 1D convolution with kernel size $k$ and stride of 3 followed by batch normalization, rectified linear unit (ReLU) activation ($f(x) = \max(0,x)$) and max pooling (which computes the maximum value in each local receptive field to downsample the feature maps) with a kernel size and stride of 2. Following were $l$ residual layers, with each layer $i$ ($i = 0,...,l-1$) containing $\mathbf{b} = \{b_i\}$ residual blocks using $\mathbf{c} = \{c_i\}$ channels. Residual blocks were either *bottleneck* or *basic* types, implemented as described in He et al.[13].

To determine the optimal values of $k$, $l$, $\mathbf{b}$, $\mathbf{c}$, and block type, the ResNet-34 and ResNet-50 architectures[13] were tested, along with variants of these with fewer channels per layer, to reduce overfitting. The SquiggleNet architecture[12] was also tested in its original form before each hyperparameter was systematically varied to find the optimal configuration for this new application. The basic block outperformed the bottleneck block and networks with a more gradual increase to a larger number of channels converged to a better loss minimum. To test the boundaries of this observation, $b$ was reduced to 1 for every layer, and the channels were configured such that $c_0 = 20$, $c_i = \lfloor c_{i-1}*1.5 \rfloor$ and $l = 10$, which was the maximum number of layers possible before the feature vectors became smaller than the receptive field. As in SquiggleNet, the kernel size was set to $k = 19$. It was found that this configuration achieved the highest accuracy on the validation set, trained using a batch size of 32 and an initial learning rate of 0.001.

**Temporal convolutional network (TCN) hyperparameter optimization.** Designed specifically for sequence modeling, TCNs[14] operate on input sequences using dilated causal convolutions; where causality is used to ensure predictions are based only on past information, while dilation allows the receptive field (RF) size to increase exponentially with network depth. When the network is sized appropriately, the last timestep in the final layer has the entire input sequence as its RF. Thus, classification predictions can be made using the last value in each channel. Residual connections are also employed to increase the depth and, hence, memory of the network. Bai et al.[14] showed the TCN is more efficient and has greater memory than equivalent-capacity recurrent networks[14].

Twenty-three TCN models were tested following the architecture described by Bai et al.[14] to identify the optimal hyperparameter configuration under the constraint that the last layer's RF covered the entire input length. As such, the number of layers $l$, kernel size $k$, and dilation base $d$ were varied such that:

$$\text{RF} = 1 + 2\sum_{i=1}^{l} 2^d(k - 1) \geq 12{,}048 \qquad (1)$$

The number of channels per layer $c$ was also varied, with the observation that more channels significantly increased training time and network size and so for practical reasons was set at or below 256. Dropout was used to regularize the network and was another hyperparameter $r$ that was optimized. The best model had parameters $l = 10$, $k = 11$, $d = 2$, $c = 32$ and $r = 0.05$, trained using a batch size of 32 and an initial learning rate of 0.0001.

**Vanilla convolutional neural network (CNN) hyperparameter optimization.** Twenty-six vanilla CNNs were also tested, hypothesizing that a simpler architecture may be more efficient yet still accurate. Each model was a variation of the following architecture: the input vector was fed into $l$ convolutional layers, each of which was composed of $b$ blocks of a 1D convolution with a stride of 1 and kernel size $k$ followed by ReLU activation. Each layer ended with a max pooling layer with a kernel size and stride of 2. The number of channels $c_i$ in layer $i$ ($i = 0,...,l-1$) was also configured, increasing with network depth to capture higher-level, more complex features. The extracted features were then passed to a

classifier $f$, which was either a simple 2-layer fully connected network with ReLU activation, a global average pooling (GAP) layer or global average pooling followed by a fully connected layer (GAP_FC). The model with the highest accuracy on the validation set had the parameters $l = 12$, $b = 1$, $k = 3$, $f$ = GAP_FC and $c_O = 20$, $c_i = \lfloor c_{i-1}*1.5 \rfloor$ and was trained using a batch size of 32 and initial learning rate of 0.0001.

**Evaluation of candidate models.** The ResNet, TCN, and CNN models with the highest accuracy on the validation set were then evaluated on the reserved testing set, which comprised all test reads from the model development datasets. The performance metrics used were accuracy (percentage of correct predictions), true positive rate (TPR) (fraction of the positive class predicted correctly), false positive rate (FPR) (fraction of the negative class predicted incorrectly), precision (fraction of correct positive predictions) and area under the receiver operating characteristic (AUROC) (a value between 0 and 1 corresponding to the area under the curve formed by plotting the TPR and FPR values across all classification thresholds). The ratio of true to false positive rate was also computed, as it better indicates the simultaneous maximization of accepted on-target molecules and rejected off-target molecules compared to the individual TPR, FPR, and precision metrics. Additionally, the mean prediction time per batch of test data ($b = 32$) was also calculated. Each model was evaluated for each of the candidate input signal lengths of {1–4} s, except for the CNN, which could not handle an input signal length of 1 s. Testing was conducted on a computer with 12 CPUs (Intel® Xeon® Platinum 8268) and one CUDA-capable GPU (NVIDIA® Tesla® V100).

### RISER software design
**Software overview.** The RISER code is comprised of independent software components responsible for data preprocessing, ReadUntil API access, model prediction, and enrichment logic to facilitate ease of maintaining, modifying, or extending the code for different applications (Suppl. Figs. 12 and 13).

**Integration with the ReadUntil API.** The ONT ReadUntil API provides an interface to access each pore in the sequencing hardware, allowing the user to request raw current data or reverse the pore voltage during sequencing to reject a molecule. Data is streamed in chunks of 1 s by default. For each signal received from the API, RISER trims the first portion of the signal corresponding to the sequencing adapter and 3′ poly(A) tail (described below). If the remaining signal is at least 2 s long (the minimum length required for the CNN) and up to 4 s long (the maximum RISER input length identified above), it is then normalized (as described above for training data preparation) and input to the CNN, which outputs a probability between 0 and 1 indicating whether the signal corresponds to the target RNA class (0: low probability, 1: high probability). If the user has requested depletion of the target class, then RISER submits a reject (*unblock* in the ReadUntil API) request when the target class probability exceeds a confidence threshold $T = 0.9$ (optimization of $T$ is described below) and allows the RNA to complete sequencing when the target class probability is less than $1 - T$. Conversely, if the user has requested enrichment of the target class, then RISER allows the RNA to complete sequencing when the target class probability exceeds $T$ and submits a reject request when the target class probability is less than $1 - T$. If the prediction is insufficiently confident, RISER will attempt to classify the molecule again once a longer signal has been received from the API.

To minimize the risk of pore damage, RISER only makes a reject request a maximum of one time per molecule. If a reject request fails (e.g., due to secondary structure formation on the trans-side of the pore that blocks ejection), it is preferable to allow the molecule to complete translocation in the forward direction. This avoids repeated futile ejection attempts that may potentially damage the pore. Finally, the ReadUntil API is encapsulated by a wrapper class in the RISER

software so that if ONT updates or replaces the API, the potentially affected code is isolated and easy to update.

**Classification threshold optimization.** To ensure RISER only makes high-confidence predictions, a classification threshold $T$ is applied when translating the probability output by the RISER model into a possible accept or reject decision (described above). To optimize $T$, three candidate thresholds (0.6: low, 0.75: medium, and 0.9: high) were tested in a controlled setting. To simulate RISER's assessment of signals that are streamed in 1 s increments from the ReadUntil API, each signal in the reserved testing set was processed into 2 s, 3 s, and 4 s lengths. The RISER model first predicted on the 2 s input, then if the prediction did not exceed the classification threshold, RISER next predicted on the 3 s and then on the maximal 4 s signal if needed. Test signals were preprocessed using BoostNano[32] (commit 8715800)[33] to remove the sequencing adapter and poly(A) tail prior to prediction. The proportion of signals with a correct prediction or no prediction was then computed for each candidate threshold (Suppl. Fig. 3), and the high classification threshold was selected to implement in RISER since it resulted in the highest proportion of correct predictions for all tested models, despite a negligible increase in the proportion of molecules that were undecided.

**Trimming the sequencing adapter and poly(A) tail.** To trim the sequencing adapter and variable-length poly(A) tail from the start of individual signals during live runs, an efficient strategy was developed that could be deployed in real-time. This was necessary as BoostNano[32] and Tailfindr[37] are incompatible with time-critical applications, and so could not be integrated with the RISER software. Both require fast5 files of complete signals as inputs rather than in-progress sequencing signals. Further, both are prohibitively slow to be used in real-time, with BoostNano utilizing an HMM, while Tailfindr uses a two-pass signal processing approach to perform a high-resolution identification of poly(A) boundaries. Conversely, speed is crucial for RISER, while imprecision in the trim position can be tolerated due to the convolutional architecture of the RISER model. Since the convolution operation is translation-invariant, the relevant components of the input signal will be recognized by the feature maps if they are present anywhere along the signal, regardless of their absolute position.

The trimming strategy developed for RISER was based on the observation that homopolymer stretches, and specifically poly(A) stretches, generate DRS signals of low variance[37]. As such, the boundaries of poly(A) stretches are delineated by a preceding sequencing adapter signal of a higher variance and lower mean current level and a subsequent transcript signal with higher variance (Suppl. Fig. 2a). These signal characteristics were exploited to identify the poly(A) tail boundaries by testing consecutive signal windows of fixed width R (Suppl. Fig. 2b). The poly(A) tail start criteria tested was: (1) at least a 20% increase in mean amplitude compared to the previous two windows, and (2) median absolute deviation (MAD) less than a threshold ($T$) (Suppl. Fig. 2b). The poly(A) tail end (trim position) criteria tested was: (1) the poly(A) tail start is found, and (2) MAD greater than 20 (Suppl. Fig. 2b). MAD was chosen instead of variance to be robust to potential outliers in the poly(A) tail current levels. Four candidate trim configurations were tested, and RISER's performance was compared with the exact trim length computed by BoostNano[32] (commit 8715800) per signal (Suppl. Fig. 2c). The configuration selected to implement in RISER was $R = 500$ signal values, which corresponds to ~10 nt for the R9.4.1 pore, and $T = 20$.

If the trim position is not identified by the above strategy within 6.2 s, then a fixed trim length of 2.2 s is used. To select the fixed trim length, the distribution of the sequencing adapter and poly(A) tail signal lengths in the model development dataset was computed (Suppl. Fig. 2d). Since it is important that the transcript signal is present in the input to the RISER model, and the presence of some poly(A)

signal is inconsequential due to the translation-invariance of RISER's convolutional architecture, the Q1 quartile (2.2 s) was selected as the fixed trim length. 6.2 s was selected as the boundary for using a fixed cutoff approach since it represents the fixed trim length (2.2 s) plus the RISER input length (4 s). Importantly, the model development dataset included reads from both natively and synthetically polyadenylated RNAs, so the selected trim length is useful regardless of the sample preparation method.

**Command-line tool.** A simple command-line tool was developed to run RISER (Suppl. Note 1). This should be executed on the same computer as MinKNOW during a sequencing run. The user must specify the mode (either *enrich* or *deplete*), the RNA class(es) to target (currently one or more of mRNA, mtRNA, globin), as well as the duration to run RISER for, which will generally be less than or equal to the MinKNOW run length. If RISER stops earlier, the MinKNOW run will continue without targeting any RNA class. The RISER run can also be set to stop later. While this does not cause an error, RISER will not receive any data during this additional time. After MinKNOW finishes the first pore scan, RISER should be started. Advanced users also have the option of specifying their own model or classification threshold.

While RISER is running, it will output real-time progress updates to the console window from which it was run (Suppl. Note 2). This includes a summary of the settings used, as well as a summary of the sequencing decisions made by RISER. A more detailed version of this information is written in a log file (Suppl. Note 3). In addition, a.csv file will be generated that lists, for each read, the batch in which it was retrieved from the ReadUntil API, the read's sequencing channel, the signal length assessed, the probability that RISER predicted for the target class(es), the classification threshold and the final accept or reject decision made (Suppl. Table 7).

**Testing RISER in a simulated live-sequencing environment.** A bulk fast5 file from a sequencing run of poly(A)+-selected RNA from an REH cancer cell line was used for testing using the MinKNOW playback tool that simulates a sequencing run. Using the default MinKNOW (v21.11.9, core v4.5.4) run settings, the bulk file was replayed 3 times for 6 h per condition: (1) without RISER (as a control), (2) with RISER targeting the mRNA class for enrichment, and (3) with RISER targeting the non-mRNA class for enrichment. Testing was performed on a desktop computer running Ubuntu 18.04 with one NVIDIA® GeForce® GTX 1650 GPU and python v3.6.9.

The sequenced reads were basecalled, mapped, and filtered using the GENCODE reference transcriptome (release 34, assembly GRCh38.p13) as described above. For each of the RISER runs, the distributions of read lengths for mapped mRNA and non-mRNA RNAs were compared using a Wilcoxon rank sum test with continuity correction ($H_1$: on-target > off-target). Reads that mapped to the GEN-CODE *protein-coding* biotype were considered mRNA, while reads that mapped to the GENCODE biotypes *Mt_rRNA*, *Mt_tRNA*, *miRNA*, *misc_RNA*, *rRNA*, *scRNA*, *snRNA*, *snoRNA*, *ribozyme*, *sRNA*, *scaRNA,* and *lncRNA* were considered non-mRNA.

**Comparison with sequence-based adaptive sampling**
**Target identification speed.** To compare the speed of RISER and sequence-based adaptive sampling (AS), 1000 reads with signals at least 6.2 s long were randomly sampled from the mRNA model test set (below). For a fair comparison, all signals were trimmed to a length of 6.2 s (the maximum signal length that RISER assesses for poly(A) trimming) so that both RISER and AS were operating on signals of equal length. RISER was configured to identify mRNA using a single GPU. For AS, each signal was basecalled and mapped by the Dorado basecalling server[16] using a single GPU for basecalling with the *RNA R9.4.1 fast* configuration. The target reference was curated by filtering the GENCODE reference transcriptome (release 34, assembly

GRCh38.p13) to retain all *protein_coding* transcripts. The time taken for RISER and AS to classify each read as belonging or not to the target mRNA class was measured. The same test was repeated but only using a single CPU.

**Input signal length needed for target identification.** To assess the input signal length that RISER and AS each need to identify targets, 1000 reads that were confidently assigned to mRNA transcripts were randomly sampled from the mRNA model test set (below). AS was performed using the same target reference and basecalling configuration as above. Both AS and RISER were run using a GPU. Starting from an input length of 2 s, AS and RISER each attempted to identify the target, with the input increased by 1 s at a time until a prediction was made. For AS, the stopping condition was the basecalled signal being mapped to one of the mRNA sequences in the target reference, while for RISER, the stopping condition was predicting the mRNA class with a probability greater than 0.9. To remove any effects due to the differences in RISER and AS's internal polyA trimming methods, the test signals were pre-trimmed using BoostNano[32].

## mRNA and mtRNA model development

**DRS datasets used.** For the development of the mRNA and mtRNA models, MinION DRS signals from human heart as well as GM12878, GM24385, HEK293, HeLa, KOPN8 and REH cell lines (sources and catalog numbers listed below) were used for training and evaluating the mRNA and mtRNA models (Suppl. Table 6). The total RNA of the GM24385 cell line was in vitro polyadenylated (as described above – RISER model development) prior to sequencing to include RNAs that are not natively polyadenylated in the training data.

**HEK293, HeLa, GM12878, KOPN8, and REH RNA extraction and sequencing.** HEK293 (cat. No. CRL-1573) and HeLa cells (human cervical cancer) (cat. No. CCL-2) were purchased from the American Type Culture Collection (ATCC) and confirmed via short tandem repeat (STR) profiling with CellBank Australia. The immortalized human peripheral vein-derived B-cells GM12878 were obtained from the Coriell Institute, while B-cell acute lymphocytic leukemia (B-ALL)-derived REH (cat. No. ACC 22) and KOPN8 (cat. No. ACC 552) cell lines were obtained from the DSMZ.

The cells were grown in DMEM medium (Gibco) supplemented with 10% fetal bovine serum (FBS). For HEK293 and HeLa cells, 1× antibiotic-antimycotic solution (Sigma) was also used. Cells were cultured in a humidity-controlled incubator at 37 °C, 5% carbon dioxide ($CO_2$). The media was replaced every 72 h, passaging and collection were done in T175 flasks at 80–90% confluency. The cells were collected from two T175 flasks when at 80–90% confluency using trypsin detachment. Five milliliters of trypsin/EDTA were used to substitute the media, and flasks were incubated at 37 °C for 5 min. The trypsinization ceased when cells were unadhered from the flasks by resuspending in 15 ml of complete DMEM media (with FBS). The suspension was then centrifuged at 200×*g* for 5 min at 4 °C. The supernatant was discarded, and cell pellets for each cell line were resuspended in 20 ml of phosphate-buffered saline (PBS) with $Ca^{2+}$/$Mg^{2+}$. The suspensions were spun down as above. The supernatant was discarded, and the PBS washing step was repeated. Residual PBS was removed, and cell pellets were stored at −80 °C and subsequently used for RNA extraction. Upon removal from −80 °C, cell pellets were thawed on ice for 10 min prior to RNA extraction. Cell pellets were then lysed in 1 ml of denaturing lysis and binding buffer (100 mM Tris-HCl pH 7.4, 1% w/v lithium dodecyl sulfate (LiDS), 0.8 M lithium chloride, 40 mM EDTA, and 8 mM DTT; LBB) by rigorous pipetting. For RNA extraction from total cell lysates, $5 × 10^6$ cells were lysed in 350 μl of RA1 lysis buffer (Macherey Nagel), and the RNA was isolated according to the manufacturer's instructions, including an on-column DNase digestion step. RNA isolated from

total cell lysate was then eluted from the columns in 80 µl of RNase-free water and stored at −80 °C.

The immortalized human peripheral vein-derived B-cells GM12878 and B-ALL-derived REH and KOPN8 cell lines were cultured with 10% FBS-supplemented RPMI1640 media (Gibco by Thermo Fisher Scientific; cat. no. 11875093) in T25 flasks (Corning; cat. no. Z717266) under 5% CO2 and 37 °C. Cells were cultured to a cell density of $1 \times 10^6$ cells/ml for REH and KOPN8 cells, while GM12878 cells were cultured to the density of $0.8 \times 10^5$ cells/ml. Following this, cells were harvested by a 5-min centrifugation at 300 rcf at 4 °C. The resultant cell pellet was washed with ice-cold PBS and separated by another centrifugation at 300 rcf for 5 min and aspirated. The protocols for subculturing, cell lysis, total RNA extraction, and direct RNA sequencing were performed as described for HEK293 and HeLa cell lines.

Polyadenylated RNA was extracted as follows. Five hundred microlitre of oligo(dT)25 magnetic beads (New England Biolabs) suspension was used per replicate for each cell pellet. The beads were washed with 1 ml of LBB twice, each time collecting the beads on a magnet and completely removing the supernatant. Upon washing, the oligo(dT)25 beads were resuspended in the pellet/LBB mixture and placed in a rotator set for 20 rpm at 25 °C for 5 min, followed by the same rotation at 4 °C for 30 min. The suspension was briefly spun down at 12,000×g, separated on a magnet, and the supernatant was discarded. The beads were then resuspended in 1 ml wash buffer (20 mM Tris-HCl pH 7.4, 0.2% v/v Titron X-100, 0.4 M lithium chloride, 10 mM EDTA, and 8 mM DTT; WB) and washed on a rotator set for 20 rpm at 4 °C for 5 min, 3 wash rounds in total. The beads were collected on a magnetic rack, and the supernatant was discarded. The wash procedure was repeated three times. The elution was carried out stepwise. The washed bead pellet was first resuspended in 50 µl of the elution buffer (25 mM HEPES-KOH, 0.1 mM EDTA; HE). The first suspension was heated at 60 °C for 5 min to facilitate the elution, and the eluate was collected upon placing the bead-sample mixture on a magnetic rack, separating the beads, and recovering the clean supernatant. The resultant pellet was next resuspended in another 50 µl of HE buffer, and the process was repeated. The eluates were then combined and subjected to an additional solid-phase reversible immobilization (SPRI) bead purification step and stored frozen at −80 °C.

RNA cleanup with SPRI beads was performed as follows. The eluate from oligo(dT) bead extraction or in vitro polyadenylated RNA was further purified using AMPure XP SPRI beads (Beckman Coulter Life Sciences) according to the manufacturer's recommendations. Briefly, the eluate samples were supplemented with 1.2× volumes of the SPRI bead suspension in its standard (supplied) binding buffer, and the resultant mixture was incubated at room temperature for 5 min with periodic mixing. The SPRI beads were brought down by a brief 2000×g spin down and separated from the solution on a magnetic rack. The supernatant was removed, and the beads were resuspended in 1 ml of 80% v/v ethanol and 20% v/v deionized water mixture and further washed by tube flipping. The bead and solution separation procedures were repeated. The ethanol-washing process was repeated one more time. Any remaining liquid was brought down by a brief spin and removed using a pipette, and the beads were allowed to air-dry while in the magnetic rack for 2 min. The purified RNA was then eluted in 20 µl of deionized water, and the RNA content was assessed using absorbance readout via Nanodrop and fluorescence-based detection via Qubit RNA high sensitivity (HS) assay kit (Thermo Fisher Scientific).

The flow cell priming and library sequencing protocol were performed as follows: 500–1000 ng of SPRI-purified RNA from each cell line was used for each 2× library preparation within every replicate (all ONT-recommended volumes doubled) with the direct RNA sequencing kit (SQK-RNA002) as supplied by ONT. SuperScript IV RNA Polymerase (Thermo Fisher Scientific) was used, RNA Control Standard (RCS) was omitted, and RNasin Plus (Promega) was included at 1 U/µl in all reaction solutions until the SPRI purification step after the reverse transcription reaction. The final adapter-ligated sample was eluted in 40 µl.

Nanopore sequencing was performed, following the same protocol as for GM24385 cells, using an ONT MinION Mk1B with R9.4.1 flow cell for 24 h. The default settings for the MinKNOW software (standalone GUI v5.7.10, core v5.7.2) were used, and the SQK-RNA002 kit was selected.

**Training data preparation.** The HeLa dataset was reserved as an independent test set, while the remaining datasets were used for retraining. Fast5 files were basecalled, mapped, and filtered using the GENCODE reference transcriptome (release 34, assembly GRCh38.p13) as described above. To only retain reads with a high-confidence biotype label, reads were discarded if their primary mapping biotype did not match the most frequent secondary mapping biotype.

For the mRNA model, the positive class comprised reads from the *protein_coding* biotype. The negative class comprised reads from the biotypes *Mt_mRNA*, *Mt_rRNA*, *Mt_tRNA*, *lncRNA*, *miRNA*, *misc_RNA*, *rRNA*, *rRNA_pseudogene*, *ribozyme*, *scRNA*, *scaRNA*, *snRNA* and *snoRNA*. Pseudogenes and the biotypes *artifact*, *non_stop_decay*, *non-sense_mediated_decay*, *processed_transcript*, *retained_intron* and *TEC* were excluded to ensure there were no common sequences between the two classes. The number of reads in the positive and negative classes were balanced to avoid model bias. Furthermore, an equal number of reads within the negative class for the biotypes *Mt_mRNA*, *Mt_rRNA*, *rRNA* and *lncRNA* was used to avoid overfitting to the more abundant mtRNA and rRNA. Balancing the sub-groups within the negative class in this manner is important to ensure strong performance for every sub-group[38,39]. To curate the set of reads for each biotype, one read from each gene was sampled at a time to minimize the overrepresentation of genes that are highly abundant in the cell lines used. Each class was split randomly into 80% training and 20% testing while preserving the biotype balance within the negative class. Ten percent of the reads in the training set were reserved for evaluating model performance after each training epoch, again preserving the balance of biotypes. For mRNA model training, 617,632 reads were used in each of the positive and negative classes.

For the mtRNA model, the positive class comprised the dominant mitochondrial RNA types in poly(A)+ DRS runs, *Mt_mRNA* and *Mt_rRNA* (Suppl. Fig. 5). The negative class comprised reads from all other biotypes, excluding *TEC* and *artifact*. The reads in the positive and negative classes were balanced. The positive class comprised equal numbers of reads from Mt_mRNA and Mt_rRNA. To ensure the more abundant Mt_mRNA genes did not dominate the positive class, the number of reads for each Mt_mRNA gene was capped at 10% of the positive class. Similarly, reads from the two Mt_rRNA genes were equally balanced. The negative class was curated using the same approach as for the mRNA model, with the biotypes *mRNA*, *rRNA* and *lncRNA* equally balanced and reads selected one at a time per gene. Preserving the biotype balance, each class was split into 80% training and 20% testing sets, with 10% of the training reads reserved for model performance evaluation during training. The training dataset consisted of 331,200 reads in each of the positive and negative classes.

The start of the raw nanopore signals was trimmed using BoostNano[32] (commit 8715800) to remove the portions of signal that correspond to the sequencing adapter and poly(A) tail, so that the models could be trained on clean transcript signals. For each signal, the first $n$ s of the remaining transcript segment of signal was then extracted and normalized using median absolute deviation with outlier smoothing. Each signal was processed three times, using $n = \{2,3,4\}$ s so that the training data contained the varying signal lengths streamed from the nanopore.

**Target-specific model training.** For each of the mRNA and mtRNA models, the RISER CNN model was trained with re-initialized weights, using binary cross-entropy loss and Adam optimization for 30 epochs. Data was fed to the network in batches of size 32, with consecutive batches randomly alternating between input lengths of 2 s, 3 s and 4 s. Accuracy was evaluated on the validation set after each training epoch, with optimum accuracies obtained after epoch 15 of 92.6% and 98.5%, respectively, for the mRNA and mtRNA models. Training was conducted using PyTorch (v1.9.0)[36] with a single NVIDIA Tesla V100 graphics processing unit (GPU).

**Testing the mRNA and mtRNA models on an independent cell line.** The performance of the mRNA and mtRNA models was evaluated using the reserved HeLa dataset (Suppl. Table 6), which was prepared using the same basecalling, mapping, filtering and biotype labeling approach described above. The resulting test set comprised 607,740 signals. BoostNano[32] was used to trim the sequencing adapter and poly(A) tail signals, to isolate the analysis to the model performance only. Performance was measured using accuracy, precision, TPR, and FPR. The accuracy of the model for each biotype within the positive and negative classes was also calculated, considering the biotypes mRNA, mtRNA, and lncRNA separately and aggregating the remaining biotypes as *other*.

**Testing the mRNA and mtRNA models on RNA sequenced by an independent laboratory.** To assess the performance of the mRNA and mtRNA models on signals acquired under different experimental conditions, they were also evaluated on an independent dataset from GM12878 cells, sequenced at The University of Birmingham as part of the Nanopore Whole Genome Sequencing Consortium[17] (Suppl. Table 6). The test dataset was prepared as described above for the independent HeLa dataset. The resulting test set comprised 506,302 signals. Performance was evaluated using the same metrics as for the independent HeLa dataset above.

**Testing RISER for the depletion of mRNA and mtRNA during live MinION sequencing**
**Sample preparation and MinION sequencing.** HeLa cells were used to evaluate mRNA and mtRNA depletion by RISER during live MinION sequencing. Cell purchase, cell culture, cell lysis, total RNA extraction and MinION sequencing were performed as described above for HeLa cells (mRNA and mtRNA model development−DRS datasets used).

**RISER usage.** RISER was run at the same time as the MinKNOW sequencing run (using the ReadUntil API v3.4.1, MinKNOW API v5.5.2, MinKNOW standalone GUI v5.7.10, MinKNOW core v5.7.2). In the RISER code, the flow cell channels were split into two groups to remove the effect of inter-flow cell variability by simultaneously testing two conditions: (1) RISER depleting both mRNA and mtRNA, and (2) no RISER (control). To split the flow cell channels, the channel numbers divisible by 2 were used for the RISER deplete condition (1) while the remaining channels were used for the control condition (2). When targeting multiple RNA classes for depletion, RISER makes a reject decision if any target class is confidently predicted (with a probability exceeding the classification threshold of 0.9), and makes an accept decision if the signal is confidently predicted to not be any of the target classes. RISER was started after the first pore scan finished and was run for the same duration as the sequencing run; 24 h. The live run was performed on a computer running Ubuntu 20.04 with one NVIDIA GeForce GTX 1650 Ti GPU and python v3.8.10.

**Globin model development**
**Whole blood collection.** Blood samples were obtained from three human donors with informed consent and approval from the ethics committee from the Australian National University (ANU) under ethics

protocol no. ETH.1.16.01/ETH.01.15.015. Whole blood was collected at the Center for Personalized Immunology (ANU) in 8.5 ml BD Vacutainer Acid Citrate Dextrose (ACD-A) Blood Collection Tubes. Two of the collected samples (1 male, 1 female) were used to produce training data for the globin model, while the remaining sample was used to evaluate the globin model in a live sequencing run.

**Total RNA extraction from whole blood.** For each of the three whole blood samples, total RNA was extracted using the PureLink RNA Mini kit (Thermo Fisher Scientific). The manufacturer's instructions were followed and 4 ml sample of blood per donor was used. On-column DNase treatment was additionally performed by adding 2 units of DNase I in its recommended buffer to the column-bound sample (New England BioLabs) and incubating at 37 °C for 15 min. The resultant RNA eluate was additionally purified using AMPure XP SPRI beads (Beckman Coulter Life Sciences) according to the manufacturer's recommendations. Briefly, the eluate RNA was supplemented with 1.2× volumes of the SPRI bead suspension and the resultant mixture incubated at room temperature for 5 min with periodic mixing. The SPRI beads were brought down by a brief 2000×*g* spin and separated from the solution on a magnetic rack. The supernatant was removed, and the beads were resuspended in 1 ml of 80% v/v ethanol, 20% v/v deionized water mixture and further washed by tube flipping. The bead and solution separation procedure and the ethanol washing process were repeated one more time. Any remaining liquid was brought down by a brief spin and removed using a pipette, and the beads were allowed to air-dry while in the magnetic rack for 2 min prior to elution. The purified RNA was then eluted in 20 μl of deionized water and the RNA content was assessed using absorbance readout via Nanodrop.

**MinION sequencing of the whole blood RNA.** For each of the two whole blood samples used for training, 200−600 ng of total RNA was used for each 2× library preparation for each sample (all recommended volumes doubled-up). Libraries were made using the direct RNA sequencing kit SQK-RNA002 (Oxford Nanopore Technologies) following manufacturer's recommendations. The final adapter-ligated sample was eluted in 20 μl (single preparation volume). The flow cell priming and library sequencing protocol were performed as described above. Nanopore sequencing was performed using an ONT MinION Mk1B equipped with R9.4.1 flow cells; samples were run for 24 h.

**Training data preparation.** For each of the two whole blood RNA sequencing runs, fast5 files were basecalled, mapped, filtered and assigned biotype labels as described above for the mRNA and mtRNA models' training data. The positive class comprised reads belonging to the globin genes *HBA1*, *HBA2*, and *HBB*, which dominate whole blood direct RNA sequencing runs. The negative class comprised reads from all other biotypes. The number of reads in the positive and negative classes were balanced. The reads in the positive class were equally balanced between *HBA(1/2)* and *HBB*, with the *HBA* portion itself equally balanced between *HBA1* and *HBA2* to ensure strong model performance for each globin gene. As in the mRNA and mtRNA models' training datasets, the negative class was curated with the biotypes *mRNA* (excluding *HBA1*, *HBA2* and *HBB* genes), *Mt_mRNA*, *Mt_rRNA* and *lncRNA* equally balanced and reads selected one at a time per gene. The negative class included some reads from the mRNA and mtRNA models' training datasets to ensure sufficient reads for each non-globin biotype. Preserving the gene balance in the positive class and the biotype balance in the negative class, each class was randomly split into 80% training and 20% testing sets, with 10% of the training reads reserved for evaluating model performance during training. 135,680 reads were used in each of the positive and negative classes in the training set. In preparation for training, the signals were processed as described above for the mRNA and mtRNA models.

**Globin model training.** To train the globin model, the RISER CNN was trained as described above for the mRNA and mtRNA models. The best accuracy on the validation set was 98.6%, obtained after epoch 6 of training.

**Testing the globin model on a reserved test set.** The performance of the globin model was evaluated on the reserved test set, which comprised 84,831 signals. The test signals were prepared, and performance was measured as described above for the mRNA and mtRNA models.

### Testing RISER for the depletion of globin mRNA during live MinION sequencing of whole blood

**MinION sequencing.** For testing globin depletion during live sequencing with RISER, the RNA extracted from the whole blood sample reserved for live testing (above) was sequenced for 24 h using the same flow cell priming, library preparation and nanopore sequencing configuration as for the mRNA and mtRNA depletion test.

**RISER usage.** RISER was run at the same time as the MinKNOW sequencing run (using the ReadUntil API v3.4.1, MinKNOW API v5.5.2, MinKNOW standalone GUI v5.7.10, MinKNOW core v5.7.2) using a split flow cell to simultaneously test (1) RISER depleting globin mRNA, and (2) no RISER (control). The same flow cell splitting procedure and computer configuration were used as for the mRNA and mtRNA depletion test.

### Evaluation of RISER's performance during live MinION sequencing runs

**Data preprocessing.** For each of the two sequencing runs used for real-time testing of RISER, the sequenced reads were basecalled, mapped and filtered using the GENCODE reference transcriptome (release 43, assembly GRCh38.p13) as described above. Per-read transcript coverage was calculated as the fraction of each transcript covered by each aligned read by parsing the alignment CIGAR string. Alignment matches (M), sequence matches (=) and sequence mismatches (X) were summed to compute the length of the aligned part of the transcript. Coverage was obtained by dividing this alignment length by the transcript length. The remainder of analyses were performed using the sequence lengths output by Guppy, the filtered alignments, the computed transcript coverage and the *decisions.csv* file output by RISER.

### Data analysis
**Read length distributions.** For each of the split flow cell conditions control (no RISER) and deplete (using RISER), the distributions of the lengths of reads assessed by RISER were compared for each RNA class using a Wilcoxon rank sum test (H₁: control > deplete), excluding outliers. The probability of superiority (PS) effect size was computed as:

$$PS = \frac{U1}{x \cdot y} \tag{2}$$

Where *U1* is the Wilcoxon rank sum test statistic, $x$ = number of reads in the control condition and $y$ = number of reads in the deplete condition.

**Per-base transcript coverage.** To plot the per-base transcript coverage for each of the example mRNA, mtRNA and lncRNA transcripts, the read depth for each split flow cell condition was computed at each reference position using samtools (v1.10) depth (with default options). The percentage depth at each reference position was calculated by dividing the read depth at that position by the total number of reads in the relevant condition along the transcript.

**Distribution of transcript fraction covered.** The transcript fraction covered by each read was computed, for each of the RNA classes, in each of the split flow cell conditions.

**Percent change in read and nt counts.** The percent change in read and nt counts in the RISER condition compared to the control condition was calculated for each transcript in each biotype. Given the objective was to deplete abundant RNA, it is critical for the most abundant RNAs within each class to be efficiently depleted. This analysis was thus restricted to the transcripts taking up at least 95% of the sequencing reads in the control condition. Given the sensitivity of percent change to low absolute read counts, transcripts with less than 30 read counts in both the RISER and control conditions were also excluded. Further, since RISER cannot distinguish lncRNAs that share sequences with mRNAs, this analysis considered lncRNAs whose exons did not overlap with protein-coding exons from the reference annotation.

To determine whether the percent change caused by RISER was statistically significant, the same calculation was performed comparing two experiments where RISER was not used. In this case, the flow cell channels of an independent run were randomly split into two groups, and the percent change in read or nt counts between the two control groups was computed for each transcript. To establish statistical significance, for each biotype, the percent changes for the set of transcripts common in both the RISER split flow cell and the control split flow cell were compared using a paired Wilcoxon signed rank test (test parameters and statistics in Suppl. Table 8).

**Available pores over time.** During each MinION run, MinKNOW conducted a pore scan every 1.5 h to assess pore health. The number of available pores found in each pore scan was extracted from the *pore_scan_data_[run_id].csv* file output by MinKNOW by counting the number of *single_pore* entries per scan for the channels in each condition. The number of available pores was then computed as a percentage of pores available in the first pore scan.

**Bias analysis.** For each RNA class, the proportion of reads correctly accepted or rejected by RISER, the proportion incorrectly accepted or rejected by RISER and the proportion for which RISER did not make a decision (i.e., because the prediction was low confidence) were calculated. The proportions of each of these outcomes were also calculated per transcript in each RNA class.

**Relative abundance.** To assess the relative abundance of transcripts in the RNA classes that were not depleted (i.e., the accepted RNA class), relative read counts were computed in each condition. For each transcript in the accepted RNA class, the relative read count was computed as the number of accepted (not rejected by RISER) reads for that transcript divided by the total number of accepted reads for all transcripts. Transcripts with less than 10 read in each of the conditions were excluded. The Pearson correlation was computed between the relative read counts in the control and RISER deplete conditions using stat_cor() in R with default parameters.

### Statistics and reproducibility
All statistical analyses performed on the data are indicated in the Methods section or figure captions. No statistical method was used to predetermine the sample size. For mRNA model development and evaluation, pseudogenes and the biotypes *artifact, non_stop_decay, nonsense_mediated_decay, processed_transcript, retained_intron,* and *TEC* were excluded from the training and testing sets to ensure there were no common sequences between the two classes. For mtRNA model development and evaluation, the biotypes *TEC* and *artifact* were excluded from the training and testing sets, as they are sequences of unknown origin. For globin model development and evaluation, the

biotypes *mRNA*, *Mt_mRNA*, *Mt_rRNA*, and *lncRNA* were used for training and testing since they represent the most abundant and thus relevant biotypes in whole blood DRS runs, so all other biotypes were excluded. Randomization was performed at the time of splitting the datasets into training, validation and testing. Randomization was also performed when determining whether the percent change in read counts per transcript caused by RISER was statistically significant, wherein the flow cell channels of an independent non-RISER run were randomly split into two groups to evaluate the percent change in read counts under control conditions. The Investigators were not blinded to allocation during experiments and outcome assessment.

### Reporting summary
Further information on research design is available in the Nature Portfolio Reporting Summary linked to this article.

## Data availability
All datasets used in this study are publicly available. The nanopore DRS signals generated in this study (GM12878-B, GM24385, HEK293-A, HEK293-C, HeLa, KOPN8, REH, and whole blood) have been deposited in the NCBI Gene Expression Omnibus (GEO) database under accession code GSE262285. The nanopore DRS signals for GM12878 cells used in this study are available in the Nanopore Whole Genome Sequencing Consortium (https://github.com/nanopore-wgs-consortium/NA12878/) Johns Hopkins University (all runs) (GM12878-A) and the University of Birmingham (run 1) (GM12878-C). The nanopore DRS signals for the human heart used in this study are available in the European Nucleotide Archive (ENA) under accession code PRJEB40410[29]. The nanopore DRS signals for HEK293 cells (HEK293-B) used in this study are available in the ENA under accession code PRJEB40872[31]. Source data are provided with this paper.

## Code availability
RISER models and code are freely available from https://github.com/comprna/riser under the GNU General Public License v3.0. A copy of the software version used for this publication is available from Zenodo[40].

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

## Acknowledgements

We thank Dr Carolina Correa Ospina and the Biomolecular Resource Facility at the John Curtin School of Medical Research, Australian National University, for the provision of an ONT MinION sequencing device for testing and for providing excellent tuition on the operation of the Min-KNOW software. For software development and testing, the National Computational Infrastructure (Australia) supercomputing resources were used through ANUMAS and NCMAS access grants. We also thank employees from Oxford Nanopore Technologies for clarification of the ReadUntil API parameters. We acknowledge funding in support of this research from a Talo Computational Biology Talent Accelerator Innovator Grant (to A.S.), Australian Research Council (ARC) Discovery Project grants DP210102385 (to E.E.) and DP220101352 (to N.H. and E.E.), Gretel and Gordon Bootes Foundation grants (to N.H. and N.S.), an Australian Government Research Training Program (RTP) scholarship (to A.S.), an Australian National University Dean's Merit Supplementary Scholarship in Science (to A.S.), and from the National Health and Medical Research Council (NHMRC) by an Ideas Grant GNT2018833 (to E.E.) and an Investigator Grant GNT1175388 (to N.S.) programs.

## Author contributions

A.S. conceived the initial idea for the project. E.E. supervised the entire project. A.S. designed and developed RISER, including data preparation, model development and evaluation, and software development and integration with the ReadUntil API. N.H. and N.S. prepared and sequenced the GM24385 cell line for training. S.J. collected the whole blood samples and S.S. contributed to the optimization of the RNA extraction from whole blood. A.R. and M.K. prepared HeLa and HEK293 cell cultures, and A.R. performed the sequencing for the RISER live sequencing experiments. N.S. supervised the RISER live sequencing experiments and advised on their design. A.S., E.E., and N.S. contributed to the interpretation of the sequencing experiment results. A.S. led the writing of the manuscript, with essential input from E.E., N.S., N.H., and A.R. All the figures in the manuscript were produced by A.S.

## Competing interests

A.S. has received funds from Oxford Nanopore Technologies for travel and accommodation to speak at the Oxford Nanopore Technologies conference London Calling 2022. The remaining authors declare no competing interests.
