## [Peer Review File · Nature Communications]

Biochemical-free enrichment or depletion of RNA classes in real-time during direct RNA sequencing with RISEREditorial Note: This manuscript has been previously reviewed at another journal that is not operating a transparent peer review scheme. This document only contains reviewer comments and rebuttal letters for versions considered at *Nature Communications*.

Reviewer #2 (Remarks to the Author):

The revisions made by the authors have substantially improved the presentation of the results. However, some major concerns persist.

Major:

1. The abstract mentions "resulting in an increase in sequencing depth of up to 93% for long non-coding RNAs". This is a misleading statement. The authors should not use a single lncRNA in Fig. 4d, that appears to be an outlier, to make such a claim. As mentioned in their letter, the average increase is 47% and this should be mentioned instead. Correspondingly the average increase of 15% for non-globin mRNA upon globin depletion should also be mentioned.
2. I do not understand the rationale and do not agree with the removal of Supplementary Table 7. That table was showing the change in read counts between control and RISER. This is absolutely critical information for the evaluation of the method and its practical application. The table needs to be converted to a barplot and provided as a main figure along Fig. 4d and 5f.
3. In response to a previous comment (reviewer 2, comment 2, round 1) regarding the effect of the method on transcriptome variance, the authors provided Sup. Fig. 9 and 13 claiming that no increase in the variance was observed. However, Fig. 5f shows an increase in the variance of non-globin mRNAs when RISER is used. This indicates that the depletion process likely introduces an imbalance in the quantification of the transcriptome. How do the authors explain this? In this context, I suggest a more careful and toned-down description when the authors propose the use of RISER in clinical settings.

Reviewer #2 (Remarks on code availability):

The code appears well maintained and designed.

Author Response to Reviewers' Comments

Round 3: Nature Communications Mar 2024

Reviewer #2

Remarks to the Author

The revisions made by the authors have substantially improved the presentation of the results. However, some major concerns persist.

We thank the reviewer for the positive feedback and have endeavoured to address all outstanding concerns, as detailed below.

Major:

1. The abstract mentions “resulting in an increase in sequencing depth of up to 93% for long non- coding RNAs”. This is a misleading statement. The authors should not use a single lncRNA in Fig. 4d, that appears to be an outlier, to make such a claim. As mentioned in their letter, the average increase is 47% and this should be mentioned instead. Correspondingly the average increase of 15% for non-globin mRNA upon globin depletion should also be mentioned.

Thanks to the reviewer for the suggestion, we agree that reporting the average metric rather than the extreme value is more important to highlight in the abstract. This has now been updated for both lncRNA and non-globin mRNA.

2. I do not understand the rationale and do not agree with the removal of Supplementary Table 7. That table was showing the change in read counts between control and RISER. This is absolutely critical information for the evaluation of the method and its practical application. The table needs to be converted to a barplot and provided as a main figure along Fig. 4d and 5f.

Supplementary Table 7 was removed from the revised manuscript since the change in read counts between control and RISER was instead added as Main Figures 4d and 5f, displayed as a boxplot showing the change in read counts between control and RISER, per transcript in each RNA class. To improve the ease of evaluation of RISER, we have now added the total read counts in each of the RISER and control conditions for lncRNA and non-globin mRNA (which is what was previously shown in Supplementary Table 7) in the legends of Figures 4d and 5f, respectively. We agree that this information is relevant to show and considered that making it available in the figure legend is sufficiently explicit and informative, considering that these are only two values.

3. In response to a previous comment (reviewer 2, comment 2, round 1) regarding the effect of the method on transcriptome variance, the authors provided Supp. Fig. 9 and 13 claiming that no increase in the variance was observed. However, Fig. 5f shows an increase in the variance of non-globin mRNAs when RISER is used. This indicates that the depletion process likely introduces an imbalance in the quantification of the transcriptome. How do the authors explain this? In this context, I suggest a more careful and toned-down description when the authors propose the use of RISER in clinical settings.

To clarify, by showing Supp. Figs 9 and 13 we are not claiming there is no variation in the relative abundance between RISER and control (non-RISER) conditions, rather that the variation in relative abundance is no greater than that observed between replicates of control experiments. This is evidenced by the Pearson correlation between relative abundances in RISER and control conditions recapitulating the Pearson correlation between relative abundances in two separate non-RISER experiments. We have updated the text to clarify this point. Moreover, as shown in Supp. Figs 9 and 13, and as it has been widely described before, the relative abundance is more variable at low expression, in both the RISER vs control comparison and the comparison between two separate non-RISER experiments. As such, with respect to relative transcript abundances, we observe the expected behaviour.

Figure 5f demonstrates a different metric to Supp. Figs. 9 and 13. Namely, Figure 5f shows, for each transcript in each RNA biotype, the percent change in read counts between RISER vs control conditions (orange) and between two control conditions (purple). Unlike relative abundance, which depends on the total number of reads in the experiment across all transcripts, the percent change depends only on the number of reads per transcript and is thus more sensitive to read count changes than relative abundance. As shown in the “between controls” comparison in Fig. 5f, in nanopore sequencing transcripts are not uniformly sampled by the pores, which means there is inherently some variation in the number of reads sequenced for each transcript, even between control conditions. It is these variations in low abundance transcripts that appear as larger percent-changes in Fig. 5f. This is not counter to, but rather consistent with, the observed levels of higher variability in relative abundance for low abundance transcripts as shown in Supp. Figs. 9 and 13. This is further supported by Supp. Fig. 12, which shows how RISER correctly accepted through the pore over 98% of non-globin mRNAs, indicating that any variability in abundances of non-globin mRNA is related to an inherent variability in sequencing rather than being introduced by RISER.

Remarks on code availability

The code appears well maintained and designed.

We thank the reviewer for the supporting comments.